# Social environment-based opportunity costs dictate when people leave social interactions
Anthony S. Gabay [1,2,3], Andrea Pisauro [4], Kathryn C. O'Nell [3] & Matthew A. J. Apps [1,2,3,5] ✉

There is an ever-increasing understanding of the cognitive mechanisms underlying how we process others' behaviours during social interactions. However, little is known about how people decide when to *leave* an interaction. Are these decisions shaped by alternatives in the environment – the opportunity-costs of connecting to other people? Here, participants chose when to leave partners who treated them with varying degrees of fairness, and connect to others, in social environments with different opportunity-costs. Across four studies we find people leave partners more quickly when opportunity-costs are high, both the average fairness of people in the environment and the effort required to connect to another partner. People's leaving times were accounted for by a fairness-adapted evidence accumulation model, and modulated by depression and loneliness scores. These findings demonstrate the computational processes underlying decisions to leave, and highlight atypical social time allocations as a marker of poor mental health.

Positive interpersonal relations are fundamental to mental health[1,2]. An abundance of research has examined the psychological processes underlying how we interact with others, highlighting that we value fairness in our social interactions. Yet much of this research ignores one of the most common responses to unfair interactions: we leave. In the real world, even if we feel unfairly treated, many reactions to others' behaviors, such as punishment, can be unwise or simply not possible. For example, rebuking a colleague who is annoying you at a party may be less common than simply walking off. These leaving decisions might be as simple as ending a phone call, leaving a conversation at work, or no longer continually responding to instant messages from a friend.

Despite conversations and many relationships, necessarily ending at some point[3], little is known about how people decide to leave or end a social interaction. Existing research has focused on the content of conversations or examined how misaligned people's preferences are for the duration of interactions[3,4]. However, few studies have examined the cognitive or computational processes underlying how we ascribe value to social interaction, evaluate that it is declining, and decide to move on from it. Strikingly, such decisions approximate problems studied in decision science and behavioural ecology about when animals stop one activity, in favour of another[5–7]. We propose that leaving a social interaction may be a similar class of decision problem, depending on similar decision-making mechanisms, and, in particular, on opportunity costs.

An opportunity cost is typically defined as the value of alternative, or foregone, activities one could be engaged in ref. 7,8. Theories of opportunity-cost processes are particularly powerful because they make specific predictions about how to optimally allocate time if you are trying to maximise a resource[5,7–11]. That is, the time you spend engaged in one activity is influenced by the value of the time you could spend in others. According to opportunity-cost theories, rich environments–where there is lots of a desired resource or where it requires little effort to find more of a resource–should make an agent move more quickly between each location or action[5,12–14].

Many behaviours from a large range of species adhere to this principle of leaving sooner from locations in rich environments when collecting rewarding resources[9–11,13,15,16]. However, in social situations other resources might be maximised over and above primary rewards. For instance, it is well established that people are highly sensitive to fairness in social interactions, foregoing rewards if they are not being equally divided between ourselves and others[17]. Here, we propose that people try to maximise average fairness across multiple social interactions. In doing so, their decisions of when to end an interaction will be shaped by how fairly the person they are inter-

[1]Centre for Human Brain Health, School of Psychology, University of Birmingham, Birmingham, UK. [2]Institute for Mental Health, School of Psychology, University of Birmingham, Birmingham, UK. [3]Department of Experimental Psychology, University of Oxford, Oxford, UK. [4]School of Psychology, University of Plymouth, Plymouth, UK. [5]Christ Church, University of Oxford, Oxford, UK. ✉e-mail: m.a.j.apps@bham.ac.uk

acting with is treating them, but also the opportunity-costs of interacting with other people in the environment. Moreover, we suggest that decisions to end social interactions will depend on evidence accumulation (EA) computational processes, based on recent evidence that such models can account for opportunity-cost-based processes[18,19].

Notably, psychiatric symptoms such as depression and loneliness are linked to less time being spent in positive social interactions, and poorer interpersonal relationships[1,2]. There is evidence that individuals who are depressed and lonely are less positive about the quality of their social interactions and report that their social environments are subjectively worse[20,21]. In addition, there is evidence that they behave differently in response to being treated unfairly[2]. However, the cognitive processes and how different patterns of social behavior from individuals who are depressed or lonely might interact with the social environments they are in are poorly understood. Here, we suggest that loneliness and depression are linked to atypical decisions of when to leave social interactions and how such decisions are shaped by the opportunity costs of social environments.

To test these hypotheses we developed a novel economic game that examines how people decide when to end social interactions that are declining in value (in terms of how fair they are being treated), in environments with different opportunity-costs. (Fig. 1; Supplementary Fig. 1). Participants were connected in one-to-one interactions with a partner who repeatedly offered them gradually declining proportions of a pot of money to share with them. The participants' choice was simply when to leave that partner and wait eight seconds to connect to a new, different partner. We manipulated the fairness of the partners–the rate at which the proportion of the pot they were sharing declined[17,20,22–24] with some decaying faster than others–similar to decays in fairness observed in multi-round economic games and real-world social interactions[25,26]. Participants made these decisions in rich or poor 'social environments', defined by the average generosity; (proportion of fair/ less fair players) in studies one, two and four, or how much effort needed to be exerted during the eight-second delay to connect to the next partner[5,10,27]. We hypothesised that people would spend less time connected to players in rich social environments, that these decisions would be best explained by an EA model that accounted for the fairness of social interactions, and that leaving times would be associated with depression and loneliness.

## Methods
This study was not pre-registered.

### Participants
175 participants (mean age = 27.6 (SD 7.6), range 18–51; 62% F) were recruited over four studies (see Supplementary Table 1 for demographics broken down by study). Gender was self-reported by participants. No data on race/ethnicity was collected. All studies were approved by The University of Oxford Central Research Committee (Studies 1–3 reference number R6061/RE001; Study 4 reference number R59122/RE001). Informed consent was obtained from all participants. In studies 1–3, participants were recruited through an Oxford University participant database. In Study 4, participants were recruited through the online platform Prolific (www.prolific.co). Participants were compensated a flat rate for their time and were also told that they would receive a bonus based on their responses to the task. However, all participants were paid the bonus, to ensure remuneration was ethical, and not based on participants being deceived by the fairness manipulations in the study. The amounts paid to participants were equal across participants in each study (£14).

### Experimental paradigm
General task structure. In our task, we had participants interact with virtual partners in different environments with unlimited potential partners. Participants' task was to decide when to leave a partner, and wait to be connected to another. We manipulated the fairness of both partners and of the environments, to examine whether people make decisions based solely on the fairness of an ongoing social interaction or

whether they also consider the opportunity cost of other potential partners. Given the consistent finding of its importance to healthy social interactions, we used fairness as a proxy for the value of interacting with a partner[17,22–24]. The general structure of the task was consistent across all four studies.

Participants were instructed that they were virtually joining different groups of people for a total of five minutes per group (Supplementary Fig. 1). Upon joining a group, they started interacting with one partner in that group until the participant decided to leave the interaction and 'travel' to interact with a different partner in the group. The interactions took the form of a repeated economic game, approximating the dictator game, with the participant in the role of receiver. Thus, every 3.5 seconds, the participant saw how their partner chose to share a new pot of credits, with varying stake sizes (Fig. 1; note that this figure displays stimulus presentation for Study 4. Studies 1–3 used text and numbers for stimulus presentation and can be seen in the Supplementary materials). All accumulated credits were added to the participant's bank, which the participants were told would be converted to a bonus payment at the end of the task.

Participants were told that decisions were collected from participants of a previous study and that those participants had been told to decide how to share money with one other person, and this would be anonymous. The participants only saw how one other individual was sharing money with them at a time. The decisions were, in fact, pre-coded by the researchers. The proportion shared by each partner decreased over time, representing a deterioration in the 'quality' of the interaction, as indexed by its fairness–defined as the proportion of the total pot shared, with 50% being completely fair and 0% completely unfair. Decays in prosocial behaviour over repeated interactions have been observed in multi-round economic games and real-world social interactions[25,26]. The participant could leave the interaction at any time by pressing the spacebar. Following a leave decision there was an eight-second delay prior to joining the next partner in the group, during which time participants received no credits, but were informed how many credits they had banked with the group so far. Participants interacted with each group for a total of five minutes. At the end of those 5 minutes they joined a completely new group of potential partners. When joining each new group, participants were explicitly told what type of group they were joining, and this information was represented throughout their time with each group by the colour of the screen's border (details for each study outlined below).

Participants were excluded from all studies if they reported awareness that the monetary splits had not come from real people. None of the participants reported such an awareness. Furthermore, it is important to note that participants who did not believe the deception would be unlikely to make any decisions to leave interacting partners, as this would lead to less monetary reward. We do acknowledge that participants may not have believed the deception without reporting this to the experimenters. However, given that these participants would likely have behaved in an economically rational manner (never to leave an interacting partner), we note that this potentially ensures that our results are robust to any biases that might have been introduced by excluding participants who didn't believe the deception (i.e., these participants would have acted counter to our hypotheses).

Crucially, while the 'fairness' of each partner's decisions decreased over the course of the interaction, the absolute value of the credits received by the participant on each trial was uniformly distributed around a constant. As such, regardless of the decay in fairness of a partner, or the fairness of others in the group, the average reward obtained was stable across sharing decisions in the experiment. The setup was such that the first offer from all other partners was always 50% plus some noise, but with no differences between groups. In terms of average reward, it was fixed such that the average reward offered to participants was the same (500 credits) every 6 trials, regardless of which environment they were in or which type of partner they were interacting with. Thus, the reward offered would always be the same on average across groups. Therefore, the economically rational behaviour in this task is to never leave a partner, unless fairness decays to 0%, as during the eight seconds it takes to connect to another partner there is no financial

reward received. Participants were informed that each partner had a finite number of decisions to make and that if the participant did not leave an interaction, they would automatically travel to the next partner when all decisions had been seen, incurring the eight-second travel cost. The number of decisions coded by each partner was 30, but participants were not instructed about this. Below, we outline task details and variations across the four studies.

**Studies 1 and 2**. The aim of study 1 was to examine whether the value of a partner and the opportunity cost of the environment influenced how long people spend in social interaction, and study 2 aimed to replicate this effect. Studies 1 and 2 used the same version of the task. The fairness of partners was manipulated by changing the rate of decay of the fairness of their sharing decisions, and the opportunity cost was manipulated by changing the proportion of fairer players (players with lower fairness

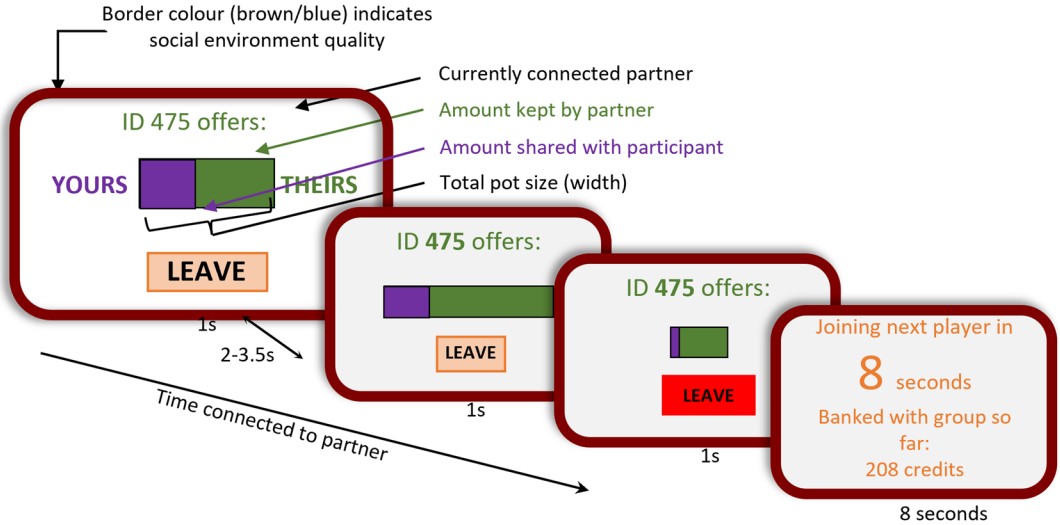

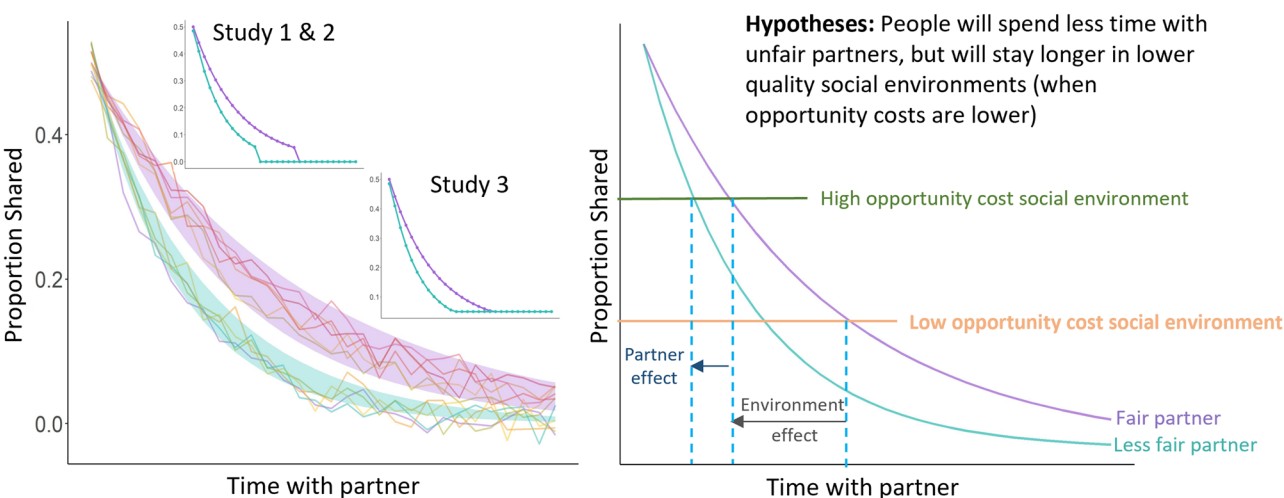

**Fig. 1 | Experimental paradigm. A** Participants were connected with partners (indicated by numeric ID numbers). Partners made decisions about how much to share out of different pots of total credits of different sizes, indicated by the width of a bar on the screen. The amount being shared was purple, and the amount kept by the partner was shown in green. Participants' task was to decide when to leave a partner to connect to another. When participants chose to leave, they experienced an eight-second delay during which they were shown the amount of credits collected in the environment so far. Participants joined different virtual "groups" of potential partners for five-minute blocks, creating different social environments. This information was indicated by a coloured border for the entirety of blocks and an instruction screen between blocks. Note that this figure shows the stimulus presentation for Study 4. Studies 1–3 used text and numerals instead as can be seen in the supplementary materials. **B** The decisions by partners were experimentally controlled with different rates of decline in fairness - the proportion shared. The lines represent examples of proportions shared by partners over time. Studies 1, 2, and 3 (inset) had only two types of partner. Study 4 had multiple partners, with noise surrounding the rates of decay of fairness. The purple shaded area represents proportions shared by "fair" partners, and the green shaded area represents proportions shared by "unfair" partners. **C** An opportunity-cost account predicts that people will leave sooner from less fair (teal line) than fair (purple) partners, but also that it will depend on the opportunity-cost afforded by the social environment. To manipulate opportunity costs for studies 1, 2, and 4, environments differed by their average generosity, determined by the proportion of fair or unfair partners included in the virtual group. In Study 3, they differed by the amount of effort (a high low number of button presses) to be completed before being connected to a new partner. When effort was low or average generosity high (green line), the opportunity cost was higher, and thus, it is predicted people would leave partners sooner than when the opportunity cost was low (orange).

decay rates) in two different social environments. A richer social environment, where the majority of partners had a low decay rate, and a poorer social environment, where the majority had a higher decay rate. Thus, studies one and two had 2 × 2 designs, in which we could examine the effect of partner fairness (fair or unfair) and social environment quality (rich or poor) on how long people spent connected to the players before deciding to leave.

The proportion shared by each partner followed an exponential decay, as shown in Eq. 1.

$$p_t = s * e^{(k*t)} \tag{1}$$

where $p_t$ is the proportion shared on trial $t$ and $k$ is the exponential decay rate. $s$, the starting value, was set to 0.5 (an equal split) for the purpose of creating the trajectory but the proportion shared of the first decision was replaced by selecting from a random uniform distribution with lower and upper bounds of 0.475 and 0.525. Partners differed in the rate of exponential decay in their levels of fairness. There were two partner types, with differing decay rates of −0.125 and −0.2, respectively (Fig. 1). We refer to partners with the lower decay rate as "fair", and those with the higher decay rate as "unfair", as this describes their behaviour overall. When the proportion dropped below 0.05, this was replaced with a zero. This has the advantage of ensuring that participants could be excluded for failing to attend, as participants should leave if they are consistently receiving none of the pot. The absolute value of each sharing decision (the credits given to the participant) was randomly sampled from a uniform distribution with lower and upper bounds of 50 and 200, with the constraint that the total sum over six decisions must equal 500. This constraint was included to ensure that, as close as possible, there would be no economically rational reason to prefer one partner over another, regardless of when one chose to leave. Stake sizes were determined by the quotient of the absolute value and proportion shared. Where these stake values equalled zero due to the fairness trajectory dropping to zero, the stake was drawn from a random uniform distribution with lower and upper bounds of 200 and 2000.

The environment manipulation in this version of the task was the average generosity of the group. When joining a new group, participants were explicitly told "Most partners shared a [high/low] proportion". Participants were instructed that the partners in each environment were randomly drawn from a large set of possible partners. The average generosity of the subset was then computed and compared to the overall average proportion shared. High generosity, therefore, meant that the average generosity of the group was greater than the average generosity overall. High generosity groups had a fair-to-unfair partner ratio of 3:1, although partners were not instructed of this ratio. Thus, in every set of four consecutive partners, participants would see three fair partners and one unfair partner (randomly ordered). The reverse was true for low generosity groups. This ensured that, as closely as possible, regardless of how many partners a participant interacted with during the five minutes, the experienced group generosity matched the 3:1 ratio of the group.

Participants joined eight groups in total, with four being high and four being low generosity, making the overall time spent on the task 40 minutes. Partner decisions were created according to the parameters described above depending on their status as fair or unfair (i.e., high or low decay in fairness), and were pseudo-randomly assigned to groups. This ensured that all participants experienced the same eight groups, but the order in which they saw the groups was fully randomized across participants.

Sharing decisions were shown in the middle of the screen (e.g., "98 out of 250") for 3.5 seconds before immediately showing the next sharing decision. When participants chose to leave one partner they were shown a countdown of eight seconds prior to connecting with the next partner.

**Study 3**. The environmental manipulation in this study was the effort required to travel between partners. Previous research has shown that repeated finger movements are considered costly, and people will avoid them, unless associated with beneficial outcomes[28,29]. Here, effort was

operationalised through repeated key presses during the 8 second delay. There were two environments, low and high effort. Effort levels were participant-specific. Prior to receiving any instructions on the task, participants were asked to press the right arrow key as many times as possible in 8 seconds. They were asked to do this three times, and the average was taken as their maximal effort. The required number of button presses was 20% and 90% of their maximum for the low and high-effort environments, respectively. This ensured that participants with greater or lesser fidelity at button presses did not consider the button presses too easy or beyond their capacity to complete. Unless otherwise stated, all other elements of this experiment were consistent with that of studies one and two.

In this study, the ratio of fair to unfair partners was 1:1, making all social environments equally generous overall. The decay rates were the same as those used for Study 1 and 2. However, in this version of the task, the proportion shared never dropped to zero. Instead, all proportions below 0.05 were replaced with 0.05. This was to remove the potential confound that any partner effect could be due to economic rationality.

Participants joined eight groups in total, with four being high, and four being low effort, making the overall time spent on the task 40 minutes. Partner decisions were created according to the parameters described above depending on their status as fair or unfair (i.e., high or low decay in fairness), and were pseudo-randomly assigned to groups. This ensured that all participants experienced the same eight groups, but the order in which they saw the groups was fully randomized across participants.

**Study 4**. The social environment manipulation in this study was average generosity, as per Studies 1 and 2. However, to examine whether results would be consistent when partners showed a much greater range in behavior, instead of having just two decay rates, the version used in this study had 17 different decay rates, and the noise was injected around each sharing decision. Decay rates for fair partners came from the set of {−0.075, −0.080, −0.085, −0.090, −0.095, −0.100, −0.105, −0.110, −0.115}, while those for unfair partners came from the set of { −0.175, −0.170, −0.165, −0.160, −0.155, −0.150, −0.145, −0.140}. The ratio of fair to unfair partners in a high generosity environment was 3:1, with every four consecutive partners containing three decay rates randomly drawn from the set of fair decay rates and one from the set of unfair decay rates. The opposite was true for low generosity environments. In this version of the task, the proportion of the first sharing decision was drawn from a random uniform distribution with lower and upper bounds of 0.47 and 0.53. Each subsequent decision was dictated by the exponential decay associated with the specific partner, but with added noise as shown in Eq. 2 below. This noise was added to each decision of each partner meaning that the trajectory of each partner's sharing decisions was unique and less predictable. All proportions below 0.05 were replaced by 0.05. The absolute value of credits shared on each decision was sampled from a random normal distribution with a mean of 400 and a standard deviation of 30.

$$p_t = s * e^{(k*t)} + \delta, \text{with } \delta \sim N(0, 0.015) \tag{2}$$

where $\delta$ is the noise term.

In this version of the task, three of the first five partners of each environment were randomly selected to include an attention check. This ensured that even those who experience fewer partners in an environment still encountered these attention checks. For these three partners, participants were asked to press a key between partner decisions once in the first four trials with that partner. They were given 2.5 seconds to do so. Failure to successfully complete an attention check held no immediate repercussions for the participant, but missed attention checks informed exclusion criteria when analysing the data.

Visual representation of the sharing decisions in this version differed to previous versions (See Fig. 1A for details). In order to reduce any potential cognitive load of calculating the proportion shared, the sharing decisions

were represented with a horizontal rectangle rather than numerically. The overall size of the rectangle represented the overall stake available to the partner for that sharing decision, and a shaded area represented the proportion of the stake shared. This rectangle appeared on the screen for 1 second, followed by a blank screen for a period of time jittered around mean 2.5 seconds, thus making the average time of each trial 3.5 seconds as per the other versions of the task.

Participants joined ten groups in total, making the overall time spent on the task 50 minutes. Partner decisions were created according to the parameters described above depending on their status as fair or unfair, and were pseudo-randomly assigned to groups. This ensured that all participants experienced the same ten groups, but the order in which they saw these groups were randomized across participants.

#### Procedure
In Studies 1–3, participants attended in-person. They were first shown visual and text instructions for the task before discussing those with the researcher to ensure comprehension. They then completed a practice run which involved spending two minutes in each type of social environment, ensuring that any potential effects of the first environment experienced were minimised[15]. Following successful completion of the task training, participants completed the real version of the task.

Study 4 recruited participants online through Prolific. In addition to the task described above, participants in this study also completed the Depression Anxiety and Stress Scales (DASS-21)[30] and the De Long Gierveld Loneliness Scale (DGLS)[1]. The order in which they completed the task and questionnaires was counterbalanced across participants. Following the task instructions, participants were asked a series of comprehension questions. If any questions were answered incorrectly twice, they were excluded from the analysis ($n = 3$). Following completion of the comprehension questions, participants completed a two-minute practice run in each type of social environment before proceeding to the task. Upon completion of the task they answered some short debrief questions.

The task was coded in PsychoPy version 2020.1.3[31], and implemented through a Windows 10 PC (Studies 1–3) or hosted on Pavlovia (pavlovia.org) (Study 4).

#### Statistical analysis
**Statistical models.** All statistical analyses were carried out using R version 4.0.2[32]. All models were analysed as linear mixed-effects models using the packages *lme4*, *lmerTest*, and *car*, with post-hoc analyses carried out with *emmeans*[33–35]. Data were visualised using the package *ggplot2* and model visualisations used the package *interactions*[36,37]. Models were compared using Bayesian Information Criterion (BIC).

To test our hypotheses, we defined the same model for Studies 1–3. The time at which the participant chose to leave an interaction ('leaving time' (LT)) was defined as a continuous outcome variable. If a participant chose not to leave a partner, the LT for that trial was defined as the onset of the final sharing decision (but see exclusion criteria below). Partner and environment types were defined as fixed-effects categorical predictors, with two levels each. We modelled both the main effects and the interaction term. We included a participant-level random intercept, as well as random slopes for each main effect (see Eq. 3). The model did not converge when including the interaction term in the random effect structure due to a lack of variance across participants. Contrasts were effect-coded such that the intercept beta represented the grand mean, and the main effect betas represented the effect averaged over the levels of the other main effect. We tested the fixed effects for statistical significance using a Type II Wald chi-square test. In the text we report the results of the chi-square test as well as the beta values and their *t*-test against zero.

$$LT \sim partner * environment + (1 + partner + environment | ID) \quad (3)$$

Due to Study 4 including multiple decay rates, the partner predictor was a continuous predictor (z-scored decay rate), rather than categorical.

The model was the same as that described above in all other respects. To test for relationships between task behavior, depression, and loneliness, we performed two additional mixed-effects models. These models were the same as for the main analysis, with the addition of depression or loneliness scores as z-scored continuous predictors (Eqs. 4 and 5, respectively). We tested for all main effects, 2-way interactions, and three-way interactions. Including depression and loneliness in a single model resulted in a worse fit (as measured by BIC) than the two being modelled separately.

$$LT \sim decay * environment * depression + (1 + decay + environment | ID) \quad (4)$$

$$LT \sim decay * environment * loneliness + (1 + decay + environment | ID) \quad (5)$$

In this task, not leaving a partner is an economically rational response. However, not actively choosing to leave a partner could also be a sign of inattention. Therefore, in Studies 1–3, we implemented an exclusion criterion that participants must have made an equal number of active decisions as there were environments. While this may remove participants from the analysis who legitimately chose to never leave a partner, it is a minimal criterion in order to ensure some engagement in the task (Study 1 excluded $n = 1$; Study 3 excluded $n = 1$). Study 4 included attention checks. Participants were excluded if they failed more than 25% of attention checks ($n = 21$). For all studies, we excluded trials where the LT was more than 2.5 standard deviations of the mean, on a within-participant, within-condition basis. Again, this was to protect against lapses in attention.

See Supplementary Methods for sample size justification.

#### Drift diffusion modelling
We modelled decisions to leave as an EA process using a drift-diffusion model (DDM)[38]. We adapted this EA process based on a recent study[18] that posited that animals make decisions to leave patches (locations) when reward foraging through an EA process that drifts towards leaving throughout a series of events of receiving rewards. This model, therefore, assumes that an animal will leave a patch when a noisy estimate of the state of the current value of rewards being obtained reaches a threshold. Here, instead of reward, we hypothesised that EA would depend on the fairness of each decision from the partner that would guide decisions of when to leave a social interaction. To test this, we deployed a model where evidence was based on the proportion shared in each decision from the partner, and compared these two other models: a standard model in which the evidence was accumulated regardless of the decisions by the partner, and a reward-based model in which the magnitude of a reward influenced a decrease in evidence to leave.

The basic process is described by the equation:

$$dEA = (k-X)dt + N(0, \sigma), with \ EA(t) \geq 0 \ \forall t \quad (6)$$

$$with \ EA(0) = \varepsilon \quad (7)$$

where $X$ is a variable that changes over the course of the accumulation. We ran three instantiations of the model which differed by the identity of the input variable $X$. $X$ could be either fairness, reward, or neither of them. The term $(k-X)$ drives the accumulation, $k$ is a parameter that modulates the input, and $N(0, \sigma)$ is a Gaussian noise term with standard deviation $\sigma$. We used $dt = 0.001$ s and assumed that the model makes a decision to leave when the evidence is greater than the threshold. i.e., $|EA| > \theta$ (the decision threshold–once reached, the participant leaves). As is common in DDMs (Ratcliff et al., 2016), we also modelled a starting bias, which in this case represents a prior bias toward leaving, $\varepsilon$. One can conceptualise this as a biassed expectation of how fair/unfair an individual might be. $\theta$, $k$, $\sigma$, and $\varepsilon$ are free parameters of the model that were fit separately to each participant's data. We ran three versions of the model where $X$ would be equal,

respectively, to:

$$X(t) = F(t) \tag{8}$$

$$X(t) = a * R(t) \tag{9}$$

$$X(t) = 0 \tag{10}$$

Where $a$ was a parameter that was scaling $R$ to match the range of values taken by $F$. As $F \epsilon (0, 1)$ it followed that $a = max(R) = 0.01$. The model with $X$ described by Eq. (11) was fitting LT distributions without any input from the task variables, offering a benchmark and baseline to compare the other models.

Each version of the model was fitted to the individual participant's leaving time (LT) data using maximum likelihood estimation. Specifically, LT distributions were computed for each participant and condition. This LT distribution was compared to the LT generated by the model through repeated simulations. For a given set of parameter estimates, we estimated the log-likelihood (LL) of the data using the following formula:

$$LL = \sum_{icond=1}^{2} \log(KS(LT_{data}^{icond}, LT_{model}^{icond})) \tag{11}$$

where KS(p,q) estimates the probability that two distributions are equal according to the Kolmogorov–Smirnov test (computed using MATLAB function *ktest2,* which in turn estimates the predicted cumulative probability through the proportions of the predicted LTs which are less than or equal to any observed LT), i_cond is an index representing the condition (the generosity of the social environment). For each participant separately, we identified the set of model parameters that maximised the LL, by searching over a grid of values: θ = {*int* 1:30}, $k$ = {0.1:1.5, in steps of 0.1}, σ = {0.1:1, in steps of 0.1} and ε = {0:0.8, in steps of 0.2}. These ranges were defined after an initial exploratory analysis over a wider range of parameter values to ensure selecting the ones that produced LT distributions spanning those seen in behaviour.

To ensure our models were robustly assessed for their ability to account for the data we (i) compared the different models computing summed BIC, (ii) compared the models in terms of the proportion of subjects for which any given model was outperforming the others (Supplementary Fig. 3), (iii) we correlated the models predicted leaving times with participants leaving times using Spearman's Rho, and (iv) We performed simulations based on the parameters estimated on subjects data. For each set of parameters, we generated LT distributions by running 1000 simulations of the model (that is, by producing this number of decision trajectories using Eq. (6) for each environment condition). To further assess the quality of the fits resulting from the best set of participant-specific parameters (those that maximised the LL function in Eq. (11)), we computed correlation coefficients between the average LT from the data and the model for all participants and conditions. We then performed the same statistics performed in the main behavioural analyses of study 4's data on the simulated data from each model. As is clear, across all these metrics only one model is highly successful in explaining behaviour. This highlights the model is a robust account of behaviour compared to a benchmark no strategy model (standard-DDM) and a model basing decisions on rewards (Reward-DDM). We did not perform a model comparison on further simplified models, as removing parameters from a DDM breaks the theoretical link between the model and the decision-making processes taking place and would, therefore, not be a principled approach.

### Reporting summary
Further information on research design is available in the Nature Portfolio Reporting Summary linked to this article.

## Results
To test an opportunity-cost-based account of how people dynamically allocate their time to interacting with different people and make decisions to leave social interactions, we conducted four studies. In each study, participants were connected to an anonymous partner and made decisions of when to leave that partner to connect to another (Fig. 1, Supplementary Fig. 1). While connected, they saw how the partner decided to share pots of money of different total magnitudes, with the fairness of these decisions–the proportion of the pot shared–decaying at different rates (creating fair and unfair partners). Importantly, pot sizes were manipulated such that, on average, all partners were equal in the total monetary value that could be obtained while being connected to them. As such, any decision to leave was economically sub-optimal, reduced the amount of money that would be earned, and thus indicated sensitivity to fairness. During the task, participants spent multiple blocks lasting five minutes interacting with partners in different social environments. Across blocks, we manipulated these environments to create high or low opportunity costs for switching between partners. An opportunity-cost based account predicts that people should spend more time with fairer partners, but should leave partners sooner when in high quality (generous or low effort) environments (Fig. 1C).

### People switch partner more frequently in more generous social environments
In Study 1 ($n = 19$) and Study 2 ($n = 25$), the social environment was manipulated by controlling the proportion with which participants would encounter two different types of partner, who differed by the rate at which the fairness of their sharing decisions decayed. There were two types of environment: high and low generosity, which differed by the ratio of fair to unfair partners (75:25 high; 25:75 low). To test our hypotheses that participants would spend less time interacting with unfair partners than fair partners, and that they would spend less time with partners in a high generosity environment than a low generosity environment, we carried out a linear mixed-effects analysis. This model comprised time spent interacting as the outcome variable, and partner-type (fair or unfair [a low or high decay in fairness of offers]), environment generosity (high or low), and their interaction, as predictors. We predicted main effects of partner-type and environment, but no interaction, in line with opportunity-cost theories[5,11].

In Study 1 (Fig. 2A), we found statistically significant main effects of partner-type and environment, with participants spending less time interacting with unfair partners than fair partners, and less time interacting with partners in a high generosity environment than in a low generosity environment (partner-type: $b = -13.86$, $SE = 1.77$, 95% CI $= -17.59$, $-10.14$; $t_{(17.88)} = -7.82$, $p < 0.001$; $X^2_{(1)} = 61.77$, $p < 0.001$; environment: $b = 3.50$, $SE = 1.53$, 95% CI $= 0.28$, $6.73$; $t_{(17.95)} = 2.28$, $p = 0.035$; $X^2_{(1)} = 5.37$, $p = 0.020$). There was also a statistically significant interaction (partner-type x environment: $b = -3.53$, $SE = 1.35$, 95% CI $= -6.17$, $-0.89$; $t_{(909.74)} = -2.26$, $p = 0.009$; $X^2_{(1)} = 6.88$, $p = 0.009$), such that the difference between high and low generosity environments was statistically significant for fair ($b = -5.27$, $SE = 1.67$, 95% CI $= -8.70$, $-1.83$; $t_{(25)} = -3.157$, $p = 0.004$), but not unfair ($b = -1.74$, $SE = 1.69$, 95% CI $= -5.20$, $1.73$; $t_{(25.9)} = -1.031$, $p = 0.3121$) partners.

Partner-type and environment effects remained statistically significant even when excluding decisions that were made after fairness had decayed to zero (Supplementary results and Supplementary Table 2). In addition, we also performed a similar mixed-effects model with the same predictors, but using the fairness (proportion shared) at the time of leaving as the outcome variable. We found the same pattern of results, with the main effects of partner-type and environment, but no interaction between them (Supplementary results and Supplementary Table 3). Notably, this behaviour was not economically rational, as leaving a partner led to an eight-second period where no reward was being collected. The patterns of behaviour outlined above, therefore, also led to people earning less money from unfair partners (by leaving them earlier) and earning less money in the high-opportunity-cost environments (Supplementary results and Supplementary Table 4).

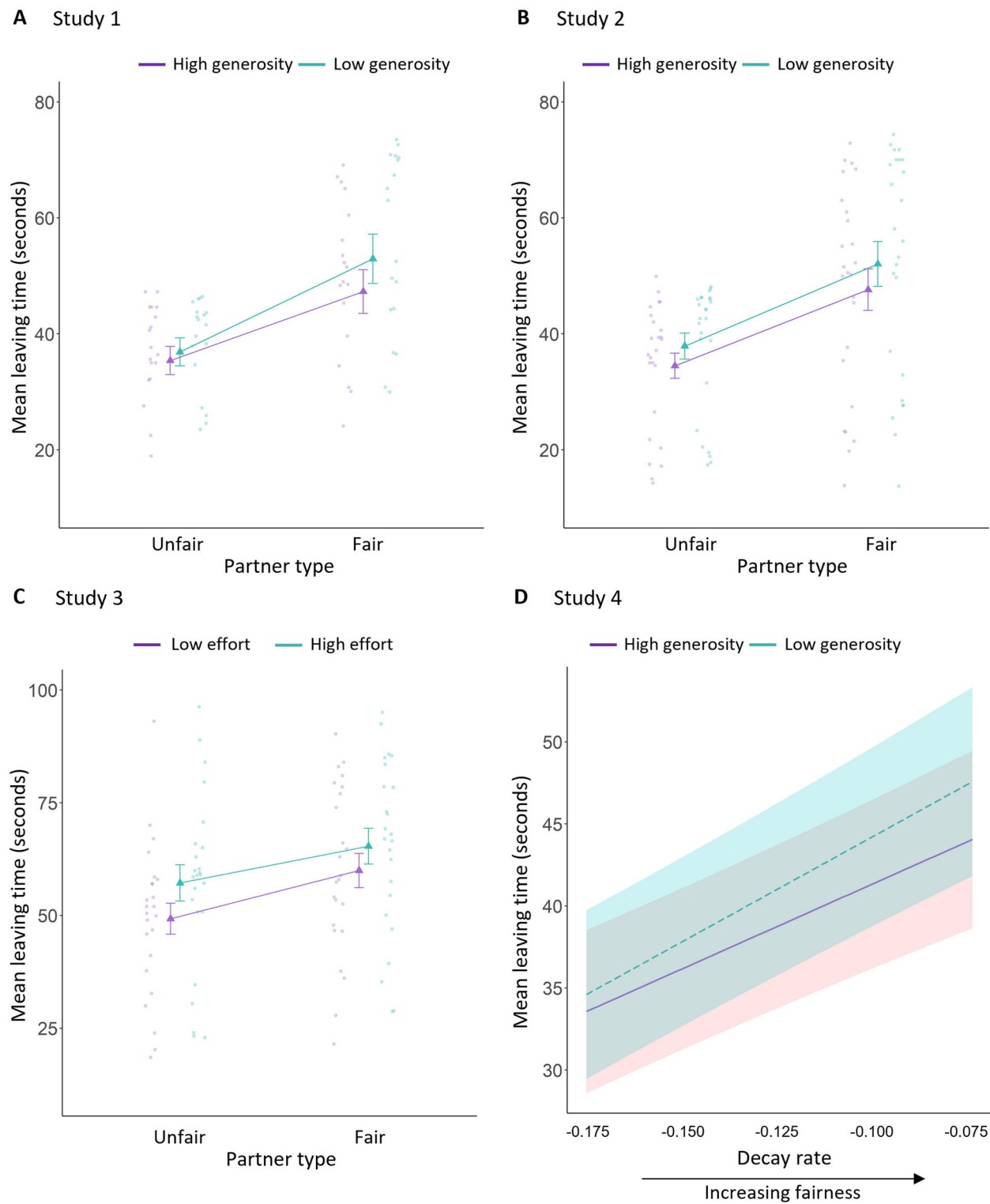

**Fig. 2 | Mean leaving times–time spent connected before deciding to leave–with partners of different types and in different quality social environments.** There were statistically significant main effects of partner-type and environment in all four studies. In all studies, participants spent more time connected with fairer partners, but less time with partners when the opportunity-cost of the environment was high (i.e., when the group was generous or when it was low effort to connect to another partner).

Means from study 1 (**A**), study 2 (**B**) where the social environment quality was defined by average generosity, high (purple) and low (teal). In study 3, **C** the social environment quality was defined by the effort–the number of button presses–required to connect to the next partner. For study 4 (**D**), partner type was a continuous variable, with shaded regions representing 95% CIs. Points represent individual participant means, and triangles represent the summary means with error bars of ±1SE.

In Study 2 (Fig. 2B), we sought to replicate these findings, using a sample size ($n = 25$) selected based on power calculations from Study 1 (Methods). We replicated the main effects of partner and social environment (partner-type: $b = -13.38$, $SE = 1.54$, 95% CI $= -16.56$, $-10.20$; $t_{(23.74)} = -8.69$, $p < 0.001$; $X^2_{(1)} = 75.48$, $p < 0.001$; environment: $b = 3.75$, $SE = 1.42$, 95% CI $= 0.82$, 6.69; $t_{(23.20)} = 2.64$, $p = 0.015$; $X^2_{(1)} = 6.97$, $p = 0.008$), but found no interaction (partner-type x environment: $b = -0.64$, $SE = 1.41$, 95% CI $= -3.41$, 2.13; $t_{(1166.58)} = -0.46$, $p = 0.649$; $X^2_{(1)} = 0.21$, $p = 0.649$). Partner-type and environment effects also remained statistically significant even if decisions after fairness had decayed to zero were excluded (Supplementary results and Supplementary Table 2). A mixed-effects model with fairness at the time of leaving as the outcome variable showed the same pattern of results, with main effects of partner type and environment (Supplementary results and Supplementary Table 3). These patterns of behaviour, therefore, also led to people earning less money from unfair partners - by leaving them earlier - and earning less money in the high-opportunity-cost environments (Supplementary results and Supplementary Table 4).

Taken together, these studies suggest that people's decisions to leave a social interaction are made based on both how fair that person is behaving, as well as the opportunity-costs of the social environment, with more generous environments favouring leaving a partner sooner.

## People switch partners more frequently when less effort is required to do so

Previous research has suggested that opportunity-costs are influenced not only by the average benefits available from alternatives in the environment, but also, the costs that must be incurred to switch from one activity to another, with higher costs leading to reduced willingness to switch[27]. This is particularly true for the effort of travelling to seek out alternatives[5,27,29,39,40]. Thus, in Study 3 we examined whether the effort required to switch from one partner to another influenced how long people spent connected, in line with an opportunity-cost account. As with Study 1 and Study 2, there were two different types of partner (fair and unfair [low or high fairness decay rate]), but in this study the ratio was 1:1 in both environments. As such, there were no differences in average generosity. Instead, the effort to connect to the next partner manipulated the opportunity-cost with participants required to exert effort during the eight-second delay after a decision to leave. This effort was operationalised as high or low numbers of repeated button presses (80% or 20% of their participants maximum, respectively). Each environment required either high or low effort.

Once again we found statistically significant effects of partner-type and environment (Fig. 2C). Participants spent less time interacting with unfair partners than fair partners (partner-type: $b = -9.10$, $SE = 1.38$, 95% CI $= -11.80$, $-6.39$; $t_{(812.58)} = -6.60$, $p < 0.001$; $X^2_{(1)} = 44.30$, $p < 0.001$), but also spent less time connected to partners when less effort was required to switch between them (environment: $b = 6.13$, $SE = 1.38$, 95% CI $= 3.43$, 8.84; $t_{(812.68)} = 4.45$, $p < 0.001$; $X^2_{(1)} = 19.53$, $p < 0.001$). There was no statistically significant interaction ($b = 1.45$, $SE = 2.75$, 95% CI $= -3.95$, 6.84; $t_{(812.32)} = 0.53$, $p = 0.599$; $X^2_{(1)} = 0.28$, $p = 0.599$). A mixed-effects model with fairness at the time of leaving as the outcome variable showed the same pattern of results, with main effects of partner-type and environment (Supplementary results and Supplementary Table 3). Thus, participants spent less time with less fair partners, but spent longer interacting with a partner if it was effortful to connect to another.

## Depressive symptoms and loneliness alter sensitivity to partner fairness and social environment quality

In Study 4, as well as replicating the effects from the previous studies there were three additional aims: (i) To better represent real-life social behaviours, we included multiple different partner types (i.e., decay rates), rather than just two (fair and unfair). We created a continuum of 17 decay rates describing the rate at which partners' sharing decisions became less fair. Additionally, we added noise around the decay curve so the change in fairness was less predictable and to make each partner's pattern of decisions

unique (Fig. 1B). (ii) to test hypotheses relating to depression and loneliness we collected a larger sample ($n = 81$) in an online version of the task, and in addition to completing the task, we had participants complete the Depression, Anxiety, and Stress Scale (DASS)[30] as well as the De Jong Gievald loneliness scale (DGLS)[1], which showed variance in depression (mean score 6.2, SD 5.9) and loneliness (mean score 3.4, SD 1.8 (supplementary Fig. 5) (iii) we also included additional blocks, in order to increase the number of samples per participant. This allowed us to fit EA models to data without over-fitting to individual data or pooling data across participants (see below).

We initially defined a mixed-effects statistical model equivalent to those in Study 1 and Study 2, except with the partner-type (fair/unfair) replaced with the continuous "decay rate" of the partner. Replicating the previous results, there was a main effect of both the decay rate of the partner and environment (partner: $b = 3.80$, $SE = 0.43$, 95% CI 2.93, 4.66; $t_{(65.13)} = 8.79$, $p < 0.001$; $X^2_{(1)} = 76.23$, $p < 0.001$; environment: $b = 2.22$, $SE = 0.82$, 95% CI 0.59, 3.86; $t_{(58.46)} = 2.72$, $p = 0.009$; $X^2_{(1)} = 7.33$, $p = 0.007$), with no statistically significant interaction ($b = 0.81$, $SE = 0.44$, 95% CI $= -0.06$, 1.67; $t_{(5362.64)} = 1.83$, $p = 0.068$; $X^2_{(1)} = 3.33$, $p = 0.068$). A mixed-effects model with fairness at the time of leaving as the outcome variable showed the same pattern of results, with statistically significant effects of partner and environment (Supplementary results and supplementary Table 3).

Next, we included scores from the DASS-21 into the mixed-effects model with leaving time as the outcome variable. The DASS-21 has three subscales (depression, anxiety and stress). Six participants did not go on to complete the questionnaires, making the sample size for this analysis n = 75. Model comparison (BIC) showed that a mixed-model with just the depression scores had a better fit than those additionally including anxiety and stress. As such the model presented here included the main effects of partner type (decay rate), environment, and depression, as well as the two-way and three-way interactions. This model showed the main effects of partner-type and environment as that presented above (partner: $b = 3.85$, $SE = 0.47$, 95% CI 2.92, 4.78; $t_{(60.02)} = 8.27$, $p < 0.001$; $X^2_{(1)} = 68.56$, $p < 0.001$; environment: $b = 2.03$, $SE = 0.85$, 95% CI 0.32, 3.75; $t_{(52.29)} = 2.38$, $p = 0.021$; $X^2_{(1)} = 5.68$, $p = 0.017$). Indluing depression in the model resulted in a change in the interaction effect, with the two-way partner-by-environment interaction becoming signifcant ($b = 0.86$, $SE = 0.46$, 95% CI $= -0.03$, 1.76; $t_{(4896.88)} = 1.89$, $p < 0.059$; $X^2_{(1)} = 4.00$, $p = 0.046$). Additionally, we found a three-way interaction between partner, environment, and depression score (Fig. 3; $b = 1.04$, $SE = 0.48$, 95% CI $= 0.10$, 1.98; $t_{(4885.67)} = 2.18$, $p = 0.030$; $X^2_{(1)} = 4.73$, $p = 0.030$). The main effect of depression, and all other two-way interactions were not statistically significant (depression, $p = 0.578$; partner-by-group, $p = 0.059$; partner-by-depression, $p = 0.208$; group-by-depression, $p = 0.986$). We found exactly the same pattern of results when including loneliness scores instead of depressions scores. There was a three-way interaction between partner, social environment, and loneliness score ($b = 0.56$, $SE = 0.26$, 95% CI $= 0.06$, 1.07; $t_{(4892.66)} = 2.18$, $p = 0.030$; $X^2_{(1)} = 4.73$, $p = 0.030$). The main effect of loneliness, and all two-way interactions were not statistically significant (loneliness, $p = 0.578$; partner-by-group, $p = 0.318$; partner-by-lonelines, $p = 0.284$; group-by-loneliness, $p = 0.607$).

Post-hoc analyses indicated that there were partner-by-depression and partner-by-loneliness interactions in the low generosity environment, but not in the high generosity environment (Fig. 3). In the low generosity environment, participants with higher depression (and loneliness scores) spent less time interacting with unfair partners than those with low depression (or loneliness) scores. This can be seen by examining the leaving times (Fig. 3) for the most and least fair partners across the range of depression/loneliness scores. Moreover, participants with higher depression and loneliness scores spent a similar amount of time interacting with unfair partners in both types of environment, which was not the case for participants with low depression scores. Notably this effect could not be explained by behaviour being more economically rational, as overall earnings for the task were not predicted

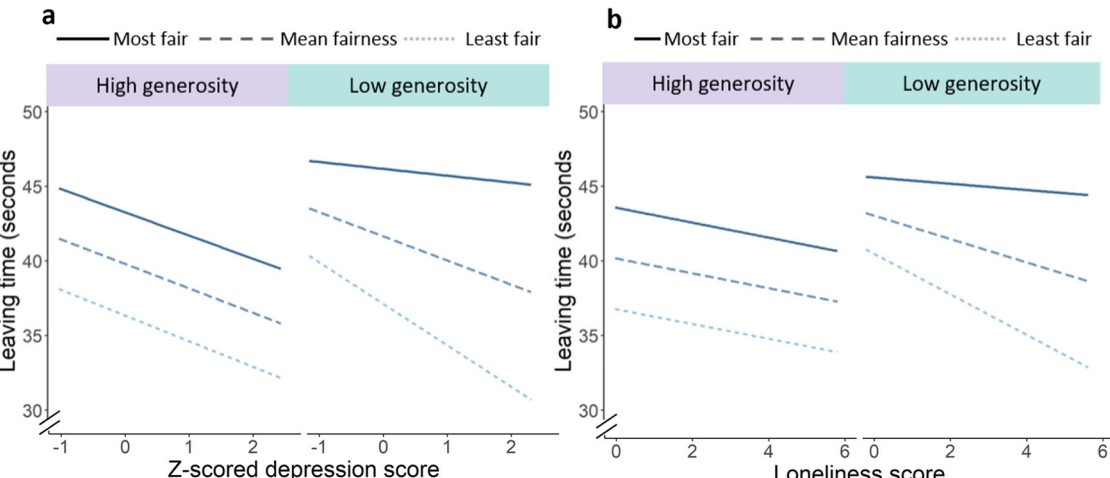

**Fig. 3 | Study 4 results. a** Depicts the statistically significant three-way interaction between partner-type, environment, and depression. **b** Depicts the three-way interaction between partner-type, environment, and loneliness. In both cases, in low generosity environments, participants with higher self-reported scores spent less time interacting with more unfair participants (three-way interaction $p = 0.030$ for both measures). Thus, higher depression and loneliness ratings were linked to different sensitivity to the fairness of partners and the quality of the social environment on decisions to leave social interactions. 'Most fair' and 'least fair' refer to mean fairness plus 1 SD and mean fairness minus 1 SD, respectively.

by depression or loneliness scores (Supplementary results and supplementary Table 5).

As such, in line with our hypotheses, these results suggest that depression and loneliness are linked to atypical decisions to leave a social interaction, and in particular to atypical integration of the fairness of someone's behaviour with information about the opportunity-costs of the social environment.

## A fairness-adapted EA model accounts for decisions to leave

To better understand the computational processes underlying how people evaluate the fairness of social interaction with information about the opportunity costs afforded by the social environment, we modeled decisions to leave partners as an EA process using an adapted DDM. This model assumed that participants are constantly and noisily accumulating evidence over the course of the interaction, and that the decision to leave is triggered when the amount of evidence reaches a threshold level (Fig. 4)[18]. The EA process is described by four parameters (see Methods for full details): (i) an initial starting value (bias); (ii) the threshold to be reached; (iii) the drift rate, which describes the rate of EA; (iv) the noise parameter, which describes how noisy the EA is. We compared the fits of a standard model (Standard-DDM), which was naive to the experimental manipulations, to two other models. In the first, the drift in EA was modulated by the fairness of each of the partner's sharing decisions (Fairness-DDM). In the second, the magnitude of the reward being shared by the partner at each decision, rather than fairness, was used to modulate the drift in EA (Reward-DDM).

We fit each of these three models to each participant's distribution of leaving times in Study 4, separately for each social environment. Model comparison (combined BIC across the two environments) revealed that the Fairness-DDM was a better fit to leaving time distributions than the Reward-DDM and the benchmark Standard-DDM (Fig. 4B). To further assess the quality of fit of the winning model, we computed correlation coefficients between the average leaving time from the data and the winning model for all participants and for each environment. The Spearman's rho was 0.98 and 0.99 for the high and low generosity environments, respectively, indicating good fit. In simulated data (Fig. 4C), the Reward-DDM failed to account for the fairness of partners, and the Standard-DDM did not predict an environment effect (Supplementary results). In contrast, the Fairness-DDM was able to capture both partner-type and environment effects present in the statistical analysis of the data above. When running a linear mixed-effects model on the simulated data from the winning Fairness-DDM, we found main effects of partner-type and social environment,

(partner-type: $b = -6.54$, $SE = 0.13$, 95% CIs = $-6.79$, $-6.28$; $t_{(161900)} = -49.827$, $p < 0.001$; $X^2_{(1)} = 3877.81$, $p < 0.001$; environment: $b = 1.66$, $SE = 0.13$, 95% CIs = 1.40, 1.91; $t_{(161900)} = 12.734$, $p < 0.001$; $X^2_{(1)} = 688.65$, $p = 0.001$; see Supplementary results and Supplementary Table 6 for the other models).

If people were making decisions to leave in line with an EA model, but adapting behaviour to the quality of the social environment, we would expect features–the parameter weights–of the model to differ between environments. In line with this, we found a higher threshold parameter in the low generosity social environment compared to the high generosity social environment ($X^2_{(1)} = 3.93$, $p = 0.048$). None of the other parameter values differed across social environments (Supplementary Fig. 4). This suggests that people required more evidence to be accumulated before making a decision to leave partners in a poor social environment when the opportunity-costs were low.

## Discussion

We often decide when to leave social interactions. This might be ending a phone call, a romantic date, or a conversation at a party. Here we tested the hypothesis that such decisions depend on how fairly we are treated, as well as the opportunity-costs of moving on to other people in the environment. Across four studies we show that people spend more time with fairer partners, but also more time with partners in poor social environments when opportunity-costs are low, either when determined by the average generosity of other people or by the effort to connect to another person. Additionally, leaving times were related to depression and loneliness, with scores on these measures linked to an interaction between how fairly a partner treated the participant and the quality of the environment. The computations underlying decisions to leave could be accounted for by an adapted DDM: the evidence to leave reflected the decaying fairness of the interacting partner, with changes in the quality of the social environment reflected by a different threshold needed to be reached before leaving. Thus, ending a social interaction may rely on EA processes that reflect the opportunity-costs of social environments.

Our results suggest that the length of time people spend in an interaction differs as a function of opportunity-costs shaped by the social environment. This was regardless of whether the opportunity-cost was manipulated through changing the average generosity of the environment, or how hard people would have to work to connect to another person. This aligns with recent research showing the importance of opportunity-costs when making other types of decisions, such as whether to exploit a current

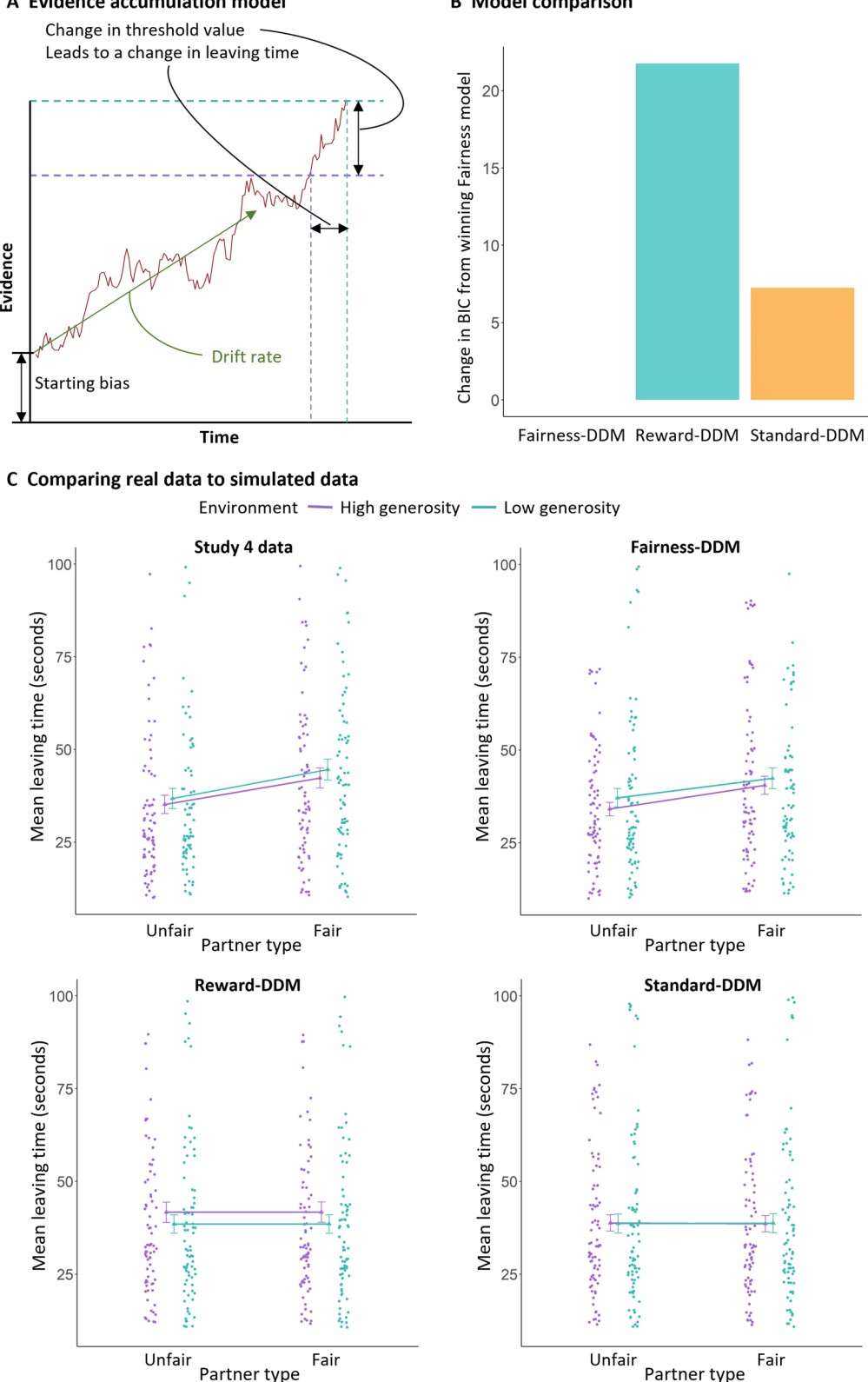

**Fig. 4 | Evidence accumulation. A** Schematic representation of evidence accumulation models. In all models evidence accumulated to leave the social interaction over time. The fairness and reward DDMs adapted this accumulation, with the drift multiplied by reward obtained (reward) or proportion or shared (fairness) at the time of each decision by the partner. **B** Model comparison of the three competing models (BIC) showed that the fairness model was best able to explain behaviour compared to the reward and standard DDMs. **C** Plots of the simulated data for each of the three models compared against data from study 4 (top left). Dots represent estimated/actual mean for a participant. Only the Fairness-DDM replicated the effects of partner-type and environment on simulated leaving times. Triangles represent estimated/actual mean across participants. Error bars represent SEM.

location or travel and explore elsewhere to obtain rewards[9,11,15]. However, unlike in other experiments, in this task participants were sacrificing rewards when making decisions to leave social interactions. Although it is possible that participants were unaware of the monetary loss involved in this study, which future research might address, this cannot explain the opportunity-cost effect of differences in leaving times between environments and is supported by evidence that people forego rewards in other studies examining fairness. This suggests that opportunity-costs apply to the value ascribed to the resource one is trying to maximise and not to reward value per se[41,42]. Here, the resource being maximised was the time spent in positive, fair interactions. Thus, our results suggest that opportunity-cost principles can be applied to a much broader range of social decisions, including how much time we allocate to social interactions[10,42]. Future research may aim to examine whether this can be applied to other social situations manipulated and studied in economic games[43], which may uncover whether leaving decisions as studied here relate to punishment or other motives that might influence decisions to stop one behaviour in favour of another.

While our studies are examining people's responses to fairness, a socially constructed phenomena, this is not to suggest that the mechanisms are necessarily socially specialised[44]. In fact, our results suggest mechanistically similar neural and computational processes for ending social interactions and other types of decision problem. Choices in the task could be accounted for by a DDM, in which evidence to leave drifted towards a threshold which triggered a decision to leave. The parameters of this model, quantifying a starting bias in evidence, a threshold boundary, the rate of evidential drift, and noise, are standard within EA models[45,46]. Such models have been used to understand the neurocognitive processes guiding perceptual, economic and social decisions, but not previously to how people decide to leave social interactions[45,47]. There were key differences in the model to standard DDMs. In particular the model was accumulating across the entire time connected to another player, but constantly changing as the fairness of the decisions of the partner changed. This model did better in a formal model comparison than a DDM in which evidence was simply accumulating over time, indicating that fairness, like reward, may be a type of evidence that can be accumulated over longer timescales[18,19]. By fitting this model separately to the different social environments, we could show a higher threshold in the poorer social environment compared to the richer i.e., when opportunity-costs were low. Taken together, these results suggest that decisions to leave a social interaction are contingent on the build-up of evidence, which depends on how much value is currently being ascribed to the social interaction. However, higher quality environments create a lower decision threshold, and thus less accumulated evidence of unfair behaviour is needed before leaving.

A plethora of existing work has shown that people value fairness, and will lose out on economic rewards, to be treated fairly[22,23]. In this respect, our participants' behaviour was consistent with previous findings. However, existing work had typically examined fairness as a static property, showing that people would punish others at a certain level of disadvantageous inequity[17,22]. Although some previous studies suggest that the context of a social interaction can influence how likely people are to punish unfair behaviour[48,49], here we have directly manipulated the opportunity-cost, examining how people decide when to leave a social interaction. In doing so, we show that fairness is not processed in an absolute manner. Instead, our results suggest that people adapt how fairly they are willing to be treated based on the average behaviour in a social environment. Indeed, we showed that people will tolerate a lower level of fairness, in an environment where most people were less generous.

Changing one's principles, such as what one considers as fair behaviour, based on the social norm, can have potentially wide social and moral consequences[50]. However, our results suggest this behaviour may be typical and adaptive when it comes to allocating time to interacting with different people. Strikingly, we found that decisions to leave social interactions correlated with depression and loneliness. Higher scores on these measures related to staying longer with fairer partners but less time with the least fair

people in the poor environment relative to the rich. As in this task the fairness of partners decayed over time, this suggests that high depression and loneliness may be linked to a tendency to react more strongly to individuals who were originally fair, but are no longer acting fairly, specifically in poor environments.

Such findings may provide a mechanism for understanding why depression and loneliness have been linked to greater sensitivity to unfair behaviour[20]. In particular our results highlight that a heightened sensitivity to unfair behaviour and being in a poor social environment may be interacting risk factors, leading to the perception of poor social relationships. In the task, by definition, every time one leaves a partner in a poor social environment they are three times more likely to encounter an unfair partner next. As such, the number of poor interactions experienced by participants with high depression or loneliness scores would be greater than those with lower scores. Although this result was exploratory and requires replication for strong conclusions to be drawn, this pattern of behaviour in everyday life might lead to a perception or belief that on average their social relationships are poorer overall than someone else in the same environment. Such subjective beliefs of poor social relationships are common in depression and chronic loneliness[20,51]. That is, this belief about their social relationships being poorer may be due to an interaction between the social environments those people are in, and potentially maladaptive behavioural responses to it (i.e., atypical decisions to end social interactions), which lead to an actually greater number of poor social interactions. The belief may therefore not be a false one, but arises not just because they are in a poor social environment, but also because of their behaviours in response to it. Thus, the interpersonal deficits linked to depression and loneliness may be better understood using decision theories that account for the dynamic allocation of time to social activities.

## Limitations

Many models of opportunity costs exist in the literature[14], and the one presented here is limited in scope in that it looks solely at how individuals choose when to leave an interaction. While being a powerful approach to understand the effect of environmental social factors on these decisions, we acknowledge that the paradigm is somewhat artificial in that participants had no choice about who to interact with in the first place. Combining these findings with an approach that models how the opportunity costs of the social environment affect initial partner selection[14,15] would provide a fuller picture of the mechanisms explored in the current manuscript.

## Conclusion

Throughout our lives, we are faced with choices of leaving social relationships, be they friendships, romantic partnerships, or simply ending a conversation. Our results show that the quality of a social environment—its opportunity-costs—are crucial for determining when people make the decisions to leave. Such processes appear normative, and guided by EA processes, with atypical consideration of opportunity-costs being a potential mechanism underlying disrupted social relationships in depression and the chronically lonely.

## Data availability
The task materials and data for this publication are accessible on the Open Science Foundation https://osf.io/urjen[52].

## Code availability
The code to reproduce the analyses for this publication is accessible on the Open Science Foundation https://osf.io/urjen[52].

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

## Acknowledgements

A.S.G was funded by an ESRC Postdoctoral Research Fellowship (ES/S011676/1). M.A.J.A. was funded by a Biosciences and Biotechnology Research Council (BBSRC) Future Leader Fellowship (BB/M013596/1) and a BBSRC David Phillips Fellowship (BB/R010668/1). K.C.O. was supported by a scholarship from the Marshall Aid Commemoration Commission. The work was also supported by the John Fell Fund and the Christ Church Research Centre. We thank Dr. Patricia Lockwood for her comments on a manuscript draft. We thank Dr Romy Froemer for useful discussions about the manuscript. The funders had no role in study design, data collection and analysis, the decision to publish, or the preparation of the manuscript.

## Author contributions

Conceptualisation by A.S.G. and M.A.J.A, who also drafted the manuscript. A.S.G, M.A.J.A, A.P., and K.C.O. edited the manuscript. Experimental design by A.S.G., M.A.J.A., and K.C.O. Data analysis by A.S.G. and A.P. Data collection by A.S.G., K.C.O. Supervision by M.A.J.A. and A.S.G. Funding acquired by M.A.J.A., A.S.G., and K.C.O.

## Competing interests

The authors declare no competing interests.
