## [Peer Review File · Communications Psychology]

20th Sep 23

Dear Dr Gabay,

We greatly apologise for the delay in processing your manuscript, which was caused by difficulties finding suitable reviewers. Thank you very much for your patience!

Your manuscript titled "Time to leave: Computations of when to end a social interaction depend on social environment-shaped opportunity-costs" has now been seen by 3 reviewers, and I include their comments at the end of this message. They find your work of interest, but raised some important points. We are interested in the possibility of publishing your study in *Communications Psychology*, but would like to consider your responses to these concerns and assess a revised manuscript before we make a final decision on publication.

We therefore invite you to revise and resubmit your manuscript, along with a point-by-point response to the reviewers. Please highlight all changes in the manuscript text file.

The reviewers provide suggestions that we hope will help you improve the clarity of your methods and strengthen the framing and discussion of your study. Although you are expected to respond to all points raised by reviewers, editorially, we ask you to pay particular attention to revising the discussion of the depression results and putting them in context of the wider literature (see Reviewer #1 and #2 for details). Related to this, please provide justification for not including anxiety and stress in your model, and also provide the raw scores for depression and loneliness.

Further, please explicitly state how the sample size was determined for each individual study.

Please also follow Reviewer #3's suggestion of conducting a formal model comparison.

For manuscripts that report null results, we require appropriate language to describe the results. (There is no statistical test that can demonstrate absence of an effect. Statements such as 'There is no difference between x and y.' or 'X does not affect Y.' must be revised to read 'We found [no/little] credible evidence of a difference between x and y.' or 'We found [no/little] credible evidence that X affects Y.')

Lastly, we encourage you to provide the full OSF link to the data and code deposition.

Please note that your revised manuscript must comply with our formatting and reporting requirements, which are summarized on the following checklist: Communications Psychology formatting checklist and also in our style and formatting guide Communications Psychology formatting guide.

Please use the following link to submit your revised manuscript, point-by-point response to the referees' comments (which should be in a separate document to any cover letter) and the

completed checklist:
[link redacted]

Please do not hesitate to contact me if you have any questions or would like to discuss these revisions further. We look forward to seeing the revised manuscript and thank you for the opportunity to review your work.

Best regards,

Antonia Eisenkoeck

Antonia Eisenkoeck
Senior Editor
Communications Psychology

EDITORIAL POLICIES AND FORMATTING

Editorial Policy: Policy requirements (Download the link to your computer as a PDF.)

* **CODE AVAILABILITY:** All Communications Psychology manuscripts must include a section titled "Code Availability" at the end of the methods section. In the event of publication, we require that

the custom analysis code supporting your conclusions is made available in a publicly accessible repository; at publication, we ask you to choose a repository that provides a DOI for the code; the link to the repository and the DOI will need to be included in the Code Availability statement. Publication as Supplementary Information will not suffice. We ask you to prepare code at this stage, to avoid delays later on in the process.

*** DATA AVAILABILITY:**

All Communications Psychology manuscripts must include a section titled "Data Availability" at the end of the Methods section or main text (if no Methods). More information on this policy, is available at <http://www.nature.com/authors/policies/data/data-availability-statements-data-citations.pdf>.

At a minimum the Data availability statement must explain how the data can be obtained and whether there are any restrictions on data sharing. Communications Psychology strongly endorses open sharing of data. If you do make your data openly available, please include in the statement:

We recommend submitting the data to discipline-specific, community-recognized repositories, where possible and a list of recommended repositories is provided at <http://www.nature.com/sdata/policies/repositories>.

If a community resource is unavailable, data can be submitted to generalist repositories such as [figshare](https://www.figshare.com/) or [Dryad Digital Repository](https://www.dryad.org/). Please provide a unique identifier for the data (for example a DOI or a permanent URL) in the data availability statement, if possible. If the repository does not provide identifiers, we encourage authors to supply the search terms that will return the data. For data that have been obtained from publicly available sources, please provide a URL and the specific data product name in the data availability statement. Data with a DOI should be further cited in the methods reference section.

REVIEWERS' EXPERTISE:

Reviewer #1: Social/economic decision making

Reviewer #2: social cognition in depression/anxiety

Reviewer #3: Social/economic decision making, computational modelling

REVIEWERS' COMMENTS:

Reviewer #1 (Remarks to the Author):

Across 4 experiments, Gabay et al. explore whether judgements about equity in social interactions adapt to the availability of equitable offers in an environment. To do this, they use a new social decision-making task which adapts a classic "patch foraging" problem from animal ecology. They find

that participants have a higher tolerance for unfair offers in environments with lots of selfish agents and for unfair offers when the effort required to find a new (potentially fairer) social exchange becomes high.

There has been a surge of interest and research in psychology and neuroscience in the last decade which has looked to apply foraging frameworks to better understand human decision making. This manuscript is – in my view - a welcome addition as it shows that the explanatory scope of the foraging framework (in particular the Marginal Value Theorem) goes beyond reward maximisation and can also be effectively applied to model social decisions and understand their individual variability (for instance in psychiatric illness).

I do have a few queries I would appreciate the authors considering:

The interpretation of the task and the results are very much framed around the task providing insight into the computations involved when in deciding about how long to make social interactions last. However, the social element here is artificially constructed (i.e., participants are not in fact interacting with real participants, this is a manipulation). Do the authors have any measure of whether participants thought the social manipulation was convincing/real?

Furthermore, there is no non-social condition included in the experiments. This would have helped make the case that what the authors are seeing here is about “positive interpersonal relations” (In 26) and relates to constructs like loneliness (Study 4 and Figure 3). Would the authors predict that participants would be more “economically rational” and be more reluctant to leave an exchange if they ran a non-social version of this task (e.g., a version in which participants were explicitly told the offers were determined by a computer algorithm)? Maybe this lack of a non social control and the implications could be discussed.

Terminology. I am not sure referring in the text to the two partners in each experiment as “fair” and “unfair” are the best terms...As I understood it, both types of partner begin offering a fair (50%) split of the pot and this decreases (i.e. could be conceived as unfair) on each trial in both cases (just at different rates).

Design. I was not clear on how the stimuli were presented to participants. In Figure 1a of the main paper, it looks like offers were presented as horizontal bars with blue / green representing the proportion for the participant and the dictator respectively. But in the Supplemental (Supp Figure 1, bottom), it seems like perhaps offers were presented as actual numbers (e.g., 97 out of 198). Can the authors please clarify this in the text (and change either figure if necessary)?

Depression/Loneliness scores: It would be beneficial to other readers to report the range, mean and sd of these questionnaire scores so they can be compared to other studies (the x axis on Figure 3 has these after zscoring so it's not possible to tell what the range is in terms of the raw scores).

Depression/Loneliness results/interpretation: The effects and interpretation of the depression/loneliness results seemed a bit weak to me. The authors report that higher scores on these measures “related to staying longer with fairer partners” [In 659/660]. This was not obvious (to me) from Figure 3? They authors report that their results suggest that “high depression and loneliness may be linked to a tendency to favour staying with individuals who were originally fair,

but are no longer acting fairly in poor environments.” [Ln 661-663]. Again – I didn’t make this connection so it would be useful to be more explicit about how the results show this. It seemed a bit of a stretch to argue in the discussion that opting to leave unfair partners quicker in a poor environment (which high depression/loneliness score participants do) could be risk factors leading to the “perception of poor social relationships” [Ln 667]. Could one not make the opposite case, that this type of behaviour avoids one being surrounded with people that treat one poorly/maladaptively (and so leads to better perception?). Finally, I was not clear why the decision to only include the depression subscale (and not anxiety or stress) in the model was made via model comparison but the decision not to include both depression and loneliness was made because of high correlation (presumably the subscales of the DASS-21 are pretty correlated also).

Reviewer #2 (Remarks to the Author):

This paper elegantly examines why individuals leave social interactions. Specifically, how individuals assign value to interactions and track their satisfaction with the interaction, and how if it is unsatisfactory they end it. They tested this by having participants engage in a novel economic game paradigm where their partner would offer them decreasing proportions of a pot of money and they tracked at what point participants left the interaction at the cost of waiting to connect to another partner.

The finding that participants forgo monetary rewards to spend less time in unfair interactions is incredibly interesting. However, it is of course unclear whether the participants knew that their decisions were not optimising their gains. Therefore, an area for future work could be to assess whether participants are explicitly aware of the cost or indeed make them aware and see if they still choose to leave.

Another area for future work would be to compare leaving choices in these interactions to other social behaviours such as retaliation commonly studied in these paradigms.

For the present study I only have minor comments. While reiteration of methodological choices can be really helpful to the reader when reading the results, they were quite extensive in the current manuscript and the authors could consider reducing overlap.

Additionally, the discussion of their depression finding was highly descriptive. The proposed mechanistic role of avoiding unfair social environments, would benefit from further elaboration and integration within the wider literature around social sensitivity and depression. Also, the authors should discuss their decision to not investigate the association with stress and anxiety.

Similarly, there was no formal power calculation for study 4. Was the study sufficiently powered to observe the three-way interaction seen in study 4?

Again, the modelling is very elegant and highly informative on what guides individuals' decisions to leave unfair social settings. I think the manuscript very significantly contributes to our understanding of social interactions.

Reviewer #3 (Remarks to the Author):

Thank you for the opportunity to review "Time to leave: Computations of when to end a social interaction depend on social environment shaped opportunity-costs". Overall, I found the study to be novel and insightful. I did wish that the authors had provided more of a theoretical link between foraging decisions and fairness to situate their studies. Some reorganizations and more details are needed to fully evaluate the impact of this paper.

Here are my comments and questions to the authors:

Introduction

1) Overall, I would urge the authors to provide a more detailed stance on their conceptualization of "leaving an interaction".

Line 32 authors state: "Yet much of this research ignores one of the most common responses to unfair interactions: we leave."

One could argue that this is a form of punishment in the social context. How is leaving different from defecting in an economic game? For the interaction partner this may represent punishment in some form. In their task that may not be the case but that depends on the cover story.

2) Lines 52-57: the authors pivot from foraging to fairness quite abruptly; I can see the similarity but they should build the point more because there are differences, not finding any more food has a different meaning for human survival than encountering unfair treatment and while there may be similarities, there may be important differences. An in depth analysis of similarities and differences would contextualize the study much better

3) Lines 66-67: Here we suggest that loneliness and depression are linked to atypical decisions of when to leave social interactions, and how such decisions are shaped by the opportunity-costs of social environments.

These are intriguing possibilities but this needs to be backed up with literature

Methods

Participants

4) I like that authors performed power analyses to determine sample sizes but what about the sample of the first study. Since they don't mention a preregistration, I am wondering about this convenience sample that is quite small and includes more females. How was this sample size determined? Also, were participants screened for clinical disorders such as anxiety and depression? Finally, I would like to see the raw scores on the loneliness and depression scales. These should be part of the demographics.

Task description:

5) Did participants see how the player shared the pot with them or between all the group members?

6) What was the rationale given to participants for why each partner tended to share less over time. This is quite artificial and may lead to some overall notions about the task being manipulated. Did authors investigate to what degree participants thought that they encountered human partners?

7) Line 151: "As such, regardless of the decay in fairness of a partner, or the fairness of others in the

group, the average reward obtained was stable across sharing decisions in the experiment” Does that refer to all initial amounts within and between groups?

8) Lines 173-187: The description would benefit from additional formulas to better understand the task structure defining stake sizes versus absolute values etc more clearly

Statistical models / Computational modeling

9) It would improve clarity to specify all linear mixed effects models

10) 332-333: There is again this one to one mapping between reward and fairness and this has not been well introduced. I understand that the computational framework for this were DDMs but I would still think that it is important to check whether they outperform simple fixed strategies or the linear mixed models. A formal model comparison should test this

11) The starting bias for the DDMs is not well explained. An additional equation might help here

12) I am missing random effects comparisons in the computational modeling section as well as parameter and model recovery.

REVIEWERS' COMMENTS:

Reviewer #1 (Remarks to the Author):

Across 4 experiments, Gabay et al. explore whether judgements about equity in social interactions adapt to the availability of equitable offers in an environment. To do this, they use a new social decision-making task which adapts a classic “patch foraging” problem from animal ecology. They find that participants have a higher tolerance for unfair offers in environments with lots of selfish agents and for unfair offers when the effort required to find a new (potentially fairer) social exchange becomes high.

There has been a surge of interest and research in psychology and neuroscience in the last decade which has looked to apply foraging frameworks to better understand human decision making. This manuscript is – in my view - a welcome addition as it shows that the explanatory scope of the foraging framework (in particular the Marginal Value Theorem) goes beyond reward maximisation and can also be effectively applied to model social decisions and understand their individual variability (for instance in psychiatric illness).

Response: We thank the reviewer for their positive appraisal of our manuscript and for their helpful comments that have allowed us to improve the work.

I do have a few queries I would appreciate the authors considering:

ø The interpretation of the task and the results are very much framed around the task providing insight into the computations involved when in deciding about how long to make social interactions last. However, the social element here is artificially constructed (i.e., participants are not in fact interacting with real participants, this is a manipulation). Do the authors have any measure of whether participants thought the social manipulation was convincing/real?

ø Furthermore, there is no non-social condition included in the experiments. This would have helped make the case that what the authors are seeing here is about “positive interpersonal relations” (In 26) and relates to constructs like loneliness (Study 4 and Figure 3). Would the authors predict that participants would be more “economically rational” and be more reluctant to leave an exchange if they ran a non-social version of this task (e.g., a version in which participants were explicitly told the offers were determined by a computer algorithm)? Maybe this lack of a non social control and the implications could be discussed.

Response: The reviewer raises important considerations here for interpretation of our results. There are two linked issues raised that are important to consider together: (i) to what extent did participants believe the experimental setup, and (ii) to what extent is the pattern of behaviour we observe necessarily strongly social or socially specialised in nature.

In regards to point (i): In the lab-based experiments participants would have been excluded if they reported awareness of the manipulation during a verbal debrief. As it was, none of the participants reported such awareness. In the online study participants were asked questions during a debrief session and participants who reported an awareness would have been excluded. As such we believe that participants who were included in analyses were not aware of the experimental manipulations. It is important to note that participants who did not believe the deception would clearly not make decisions to leave in the experiment as this would lead to less monetary reward. As such, it seems unlikely that participants would show the pattern of results we found unless they believed the experimental setup. Thus, we also note that this potentially ensures our results are robust to

any biases that might have been introduced by excluding participants who didn't believe the deception.

In regards to point (ii): We did not intend to make strong claims about this being a socially specialised set of processes (for a full discussion on our approach to considering social specialisation see Lockwood, Apps, Chang, 2020, *TICS*). Our suggestion is that computations that can be applied to reward based choice extend to fairness based choices. Indeed, it is plausible that if participants were instructed that offers were from AI agents there might be a response to the fairness of those agents (see Jones-Jang & Jin Park, *Journal of Computer-Mediated Communication*, 2022), with previous work showing that of course people are good (if not optimal) at making reward based foraging choices (Le Heron et al., 2020). As such, although the mechanisms we identify may have important impacts on social behaviour, it is not the case that they appear to be highly specialised for social information processing.

To address the reviewer's concerns we have now highlighted the exclusion criteria in the methods for participants who reported not believing the deception and the point that participants would likely make economically rational choices and not leave if that were the case (lines 194 - 202). We have also addressed additional considerations about social specialisation relevant to other reviewers points in the methods and discussion of the manuscript (lines 712 - 727).

ø Terminology. I am not sure referring in the text to the two partners in each experiment as "fair" and "unfair" are the best terms...As I understood it, both types of partner begin offering a fair (50%) split of the pot and this decreases (i.e. could be conceived as unfair) on each trial in both cases (just at different rates).

Response: We agree with the reviewer that the labels "fair" and "unfair" are perhaps not fully clear. We do think it is important in the figures for clarity that we keep the labels "fair" and "unfair" and in the text as alternative terms such as "high decay" become harder to parse in a complex results section. However, in the text we have now ensured that the conditions are labelled as different "decays in fairness" when it improves clarity to do so ad in several places, so that it is very clear that the phrase fair or unfair refers to decay rates.

ø Design. I was not clear on how the stimuli were presented to participants. In Figure 1a of the main paper, it looks like offers were presented as horizontal bars with blue / green representing the proportion for the participant and the dictator respectively. But in the Supplemental (Supp Figure 1, bottom), it seems like perhaps offers were presented as actual numbers (e.g., 97 out of 198). Can the authors please clarify this in the text (and change either figure if necessary)?

Response: We apologise to the reviewer about the lack of clarity. Studies 1-3 used a purely text based description of offers, whereas study 4 used the horizontal bars to display offers. This was to reduce any cognitive load on participants and ensure that our measures were generalisable across different stimulus presentation forms. We have now revised descriptions in the methods, figure legends and supplement to ensure this is clear.

ø Depression/Loneliness scores: It would be beneficial to other readers to report the range, mean and sd of these questionnaire scores so they can be compared to other studies (the x axis on Figure 3 has these after zscoring so it's not possible to tell what the range is in terms of the raw scores).

Response: This is a very helpful point from the reviewer, we agree it would be useful to display the distributions of these scores and we now include histograms in a supplementary figure and report the mean and SD (lines 581 - 582).

ð Depression/Loneliness results/interpretation: The effects and interpretation of the depression/loneliness results seemed a bit weak to me. The authors report that higher scores on these measures “related to staying longer with fairer partners” [In 659/660]. This was not obvious (to me) from Figure 3?

Response: We agree with the reviewer that the results of the three way interaction between partner fairness, environmental fairness and depression/loneliness could have been more clearly described. In line with other reviewers points in addition to this we have now re-worded and modified how these results are summarised and interpreted in the manuscript, although we note that as this was an exploratory analysis of a new finding we did not want to interpret the results too strongly (lines 614 - 618, paragraph beginning line 712).

They authors report that their results suggest that “high depression and loneliness may be linked to a tendency to favour staying with individuals who were originally fair, but are no longer acting fairly in poor environments.” [In 661-663]. Again – I didn’t make this connection so it would be useful to be more explicit about how the results show this. It seemed a bit of a stretch to argue in the discussion that opting to leave unfair partners quicker in a poor environment (which high depression/loneliness score participants do) could be risk factors leading to the “perception of poor social relationships” [In 667]. Could one not make the opposite case, that this type of behaviour avoids one being surrounded with people that treat one poorly/maladaptively (and so leads to better perception?).

Response: We apologise to the reviewer and agree that this wording was not particularly clear. As above, we have now significantly modified how we discuss these results in the manuscript.

Finally, I was not clear why the decision to only include the depression subscale (and not anxiety or stress) in the model was made via model comparison but the decision not to include both depression and loneliness was made because of high correlation (presumably the subscales of the DASS-21 are pretty correlated also).

Response: We apologise to the reviewer this wasn’t completely clear in the manuscript. Our selection was made based on model comparison i.e. including stress or anxiety in a model led to a higher BIC, and thus it would be invalid to include to interpret the results of models including them. In contrast, including depression or loneliness improved BIC, but including multiple measures in one model would be invalid because they are too highly correlated and the result would be difficult to interpret. We have now reported this in the manuscript more clearly (line 355, and lines 376 - 377).

Reviewer #2 (Remarks to the Author):

This paper elegantly examines why individuals leave social interactions. Specifically, how individuals assign value to interactions and track their satisfaction with the interaction, and how if it is unsatisfactory they end it. They tested this by having participants engage in a novel economic game paradigm where their partner would offer them decreasing proportions of a pot of money and they tracked at what point participants left the interaction at the cost of waiting to connect to another partner.

The finding that participants forgo monetary rewards to spend less time in unfair interactions is incredibly interesting. However, it is of course unclear whether the participants knew that their decisions were not optimising their gains. Therefore, an area for future work could be to assess whether participants are explicitly aware of the cost or indeed make them aware and see if they still choose to leave.

Another area for future work would be to compare leaving choices in these interactions to other social behaviours such as retaliation commonly studied in these paradigms.

Response: We agree with the reviewer that these are all interesting points. We think that even if participants weren't aware that it was reward maximising to stay longer, previous work suggests people do not reward maximise in other experiments such as the ultimatum game or prisoners' dilemma when it is clear that the behaviour is not reward maximising. Thus, clearly people do forgo money in the sake of fairness when it is explicitly clear to them, suggesting similar processes might happen here. We have now discussed this issue and the note of future work in the revised discussion (lines 700 - 703, and lines 708 - 711).

For the present study I only have minor comments. While reiteration of methodological choices can be really helpful to the reader when reading the results, they were quite extensive in the current manuscript and the authors could consider reducing overlap.

Response: We apologise to the reviewer for any repetition, we have removed this in the parts of the manuscript where we feel it was not useful.

Additionally, the discussion of their depression finding was highly descriptive. The proposed mechanistic role of avoiding unfair social environments, would benefit from further elaboration and integration within the wider literature around social sensitivity and depression. Also, the authors should discuss their decision to not investigate the association with stress and anxiety.

Response: We agree with the reviewer that the discussion of depression results was a little descriptive. Part of this was because it is a new finding, in a new paradigm, that was an exploratory result, so we did not want to over interpret our findings. However, all reviewers noted this point, and so we have now discussed these results in more detail in the revised discussion (paragraph beginning line 712). In addition, as noted to the previous reviewer, stress and anxiety were investigated but did not improve BIC scores, indicating they do not meaningfully associate with behaviour in the task in this sample.

Similarly, there was no formal power calculation for study 4. Was the study sufficiently powered to observe the three-way interaction seen in study 4?

Response: We thank the reviewer for noting that this wasn't outlined in the manuscript. As a novel, exploratory study, the sample size could only be based on estimates of the size of effect which were unknown, and were also considering the balance with finding a meaningful effect size and thus not having an extremely large sample. It is recommended that post-hoc power analyses are not used as an estimate of power (Dziak et al., 2020). A recommended alternative is to calculate the bootstrapped CI of the beta value of effects of interest, and if 0 is not within the interval it provides evidence that the study has sufficient power. We have now done this and zero is not within the CI around the beta. We have now reported this in the sample size justification section of the supplementary materials, to demonstrate we have significant power for our result.

Again, the modelling is very elegant and highly informative on what guides individuals' decisions to leave unfair social settings. I think the manuscript very significantly contributes to our understanding of social interactions.

Response: We thank the reviewer for their helpful review and positive appraisal of the manuscript.

Reviewer #3 (Remarks to the Author):

Thank you for the opportunity to review “Time to leave: Computations of when to end a social interaction depend on social environment shaped opportunity-costs”. Overall, I found the study to be novel and insightful. I did wish that the authors had provided more of a theoretical link between foraging decisions and fairness to situate their studies. Some reorganizations and more details are needed to fully evaluate the impact of this paper.

Response: We thank the reviewer for their helpful review, and are glad they found the manuscript novel and insightful. We hope that the revised manuscript addresses their concerns.

Here are my comments and questions to the authors:

Introduction

1) Overall, I would urge the authors to provide a more detailed stance on their conceptualization of “leaving an interaction”.

Response: We apologise to the reviewer that this wasn’t clear in the original manuscript. We conceptualise leaving an interaction as a class of decision that is distinct from other types of socio-cognitive process that has been studied experimentally or the type of decision studied in most non-social paradigms. The analogy to foraging is that in foraging research there is a distinction between binary choices (e.g. choosing between two options), “predator-prey” decisions which involve encountering a potential option and deciding whether to engage with it or travel further to find an alternative, or “patch leaving” decisions, which is to stop collecting reward in a current location and travel elsewhere. These are different classes of choice problems. We draw the analogy here to patch leaving to leaving an interaction. We use this framework and suggest it might apply to lots of decisions of when to stop a social interaction (e.g. in a face to face setting with friends or colleagues, online on social media, and perhaps even to mating choices.) The advantage of conceptualising the problem this way is that it can explain how the brain didn’t need to involve novel mechanisms to make choices to end an interaction versus ending reward foraging in a patch. We do note that future work will need to unpack what specific types of social decisions it does or doesn’t apply. In the revised introduction we believe we have made this clearer.

Line 32 authors state: “Yet much of this research ignores one of the most common responses to unfair interactions: we leave.”

One could argue that this is a form of punishment in the social context. How is leaving different from defecting in an economic game? For the interaction partner this may represent punishment in some form. In their task that may not be the case but that depends on the cover story.

Response: The reviewer is correct that leaving a social interaction could be considered a punishment in some settings, but we argue it also might not be in others. For instance, leaving an interaction with a boss who is not treating you fairly is not about punishing the

other, it is about the aversiveness of being treated unfairly. However, the aim of our manuscript was to identify whether leaving interactions changed in different environments shaped by the average fairness one could experience or the opportunity cost of other potential people to interact with. There was no opportunity to punish and the study didn't examine the self-reported motivations underlying behaviour. Future work may be able to unpick the different motives that drive such decisions, but that was beyond the scope of the questions addressed in the four studies in this manuscript. In the revised manuscript we have now outlined how there may be different motives to leaving, including punishment, and in the future work should try and unpick these motives (lines 708 - 711).

2) Lines 52-57: the authors pivot from foraging to fairness quite abruptly; I can see the similarity but they should build the point more because there are differences, not finding any more food has a different meaning for human survival than encountering unfair treatment and while there may be similarities, there may be important differences. An in depth analysis of similarities and differences would contextualize the study much better

Response: We agree with the reviewer, as noted above we draw the parallels to the patch leaving literature but this could be clearer. In patch leaving decisions are dependent upon the value of a patch and the average patch value in the environment. Our argument was that fairness is a property that people value in social interactions and thus the same principles may apply when the value isn't reward but is fairness. We have now revised the introduction to make these links clearer (paragraph beginning line 59).

3) Lines 66-67: Here we suggest that loneliness and depression are linked to atypical decisions of when to leave social interactions, and how such decisions are shaped by the opportunity-costs of social environments.

These are intriguing possibilities but this needs to be backed up with literature.

Response: The reviewer raised an important point. This was an exploratory question and the hypotheses was derived based on the fact that higher levels of loneliness and depression have been linked to different social behaviours (Hirschfeld et al., 2000, J Clin Psychiatry), different social decisions in the lab (including those linked to fairness (Kupferberg et al, 2016, Neurosci Biobehav Rev), and that people have a different perception of their social environments in the real world. Our hypothesis was therefore that levels of loneliness and depression will be associated with different patterns of behaviour in the task that manipulated these variables. However, we did not make this clear and in the revised manuscript we believe it is (paragraph beginning line 72).

Methods

Participants

4) I like that authors performed power analyses to determine sample sizes but what about the sample of the first study. Since they don't mention a preregistration, I am wondering about this convenience sample that is quite small and includes more females. How was this sample size determined? Also, were participants screened for clinical disorders such as anxiety and depression? Finally, I would like to see the raw scores on the loneliness and depression scales. These should be part of the demographics.

Response: We thank the reviewer for highlighting these omissions. The first study was a convenience sample as the reviewer suggests. The sample size was initially based on the estimate that fairness is a highly powered effect in behavioural economic games - most participants reject some offers in economic games where it is economically irrational. As such the effect size of fairness is large. However, it was unknown how strong the effect size

of the opportunity costs would be, with only a few examples even for reward based choices that had not been analysed or published at the time of conducting this study (Le Heron et al., 2020). We therefore estimated that a sample of size of 24 might demonstrate effects. We then ensured that we conducted power analyses for the two subsequent replications. We note we didn't formerly exclude people with neurological or psychiatric conditions in the lab-based studies, but these were participants from the University database which contains mostly studies. However, given the high prevalence of anxiety and depression in such samples, we did not want to exclude for fear of then examining a very unrepresentative sample. However, we feel that replicating the effects in the online study, in a prolific sample, demonstrates the results are independent of sampling procedures. We agree with regard to depression and loneliness scores, and now report means, SDs (line 581) and plotted histograms of scores in the supplementary material.

Task description:

5) Did participants see how the player shared the pot with them or between all the group members?

Response: Participants only saw what they believed were offers from one other person at a time, for how that other person had decided to split the pot of money between one other person (the participant) and what they would keep themselves. We have now re-worded the methods to make this clear (lines 178 - 181).

6) What was the rationale given to participants for why each partner tended to share less over time. This is quite artificial and may lead to some overall notions about the task being manipulated. Did authors investigate to what degree participants thought that they encountered human partners?

Response: We thank the reviewer for noting that this information was not clear in the manuscript. No rationale was given to participants. Giving a rationale for what they were instructed was the free choices of other people would have potentially confused or raised suspicions in participants that the experimental setup was artificial. Although it could be argued that it is artificial, there is evidence that people become less prosocial over choices during repeated interactions (Chaudhuri, 2011; The Handbook of Experimental Economics, 1997). So in fact it is possible that real participants would have made decisions in a somewhat similar manner.

Participants were excluded from the studies 1-3 if they reported not believing the deception in the debriefing session. It is also important to note that participants who did not believe the deception would clearly not make decisions to leave in the experiment as this would lead to less monetary reward. As such, it seems unlikely that participants would show the pattern of results we found unless they believed the experimental setup.

To address the reviewer's concerns we have now highlighted the exclusion criteria in the methods for participants who reported not believing the deception and the point that participants would likely make economically rational choices and not leave if that were the case (paragraph beginning line 194). We have also emphasised that people do show a decay in how prosocial they are in iterative economic games (lines 184-185).

7) Line 151: "As such, regardless of the decay in fairness of a partner, or the fairness of others in the group, the average reward obtained was stable across sharing decisions in the experiment" Does that refer to all initial amounts within and between groups?

Response: The reviewer raises an important point about how rewards must be equal across conditions. The setup was such that the first offer from all other partners was always 50% plus some noise, but with no differences between groups. In terms of average reward, it was fixed such that the average reward offered to participants was the same (500 credits) every 6 trials, regardless of which environment they were in or which type of partner they were interacting with. Thus, the reward offered would always be the same on average. We have now highlighted this in the methods (lines 207 - 211).

8) Lines 173-187: The description would benefit from additional formulas to better understand the task structure defining stake sizes versus absolute values etc more clearly

Response: We apologise to the reviewer but on this point we are a little confused as to what they refer to. The average reward offered to participants was fixed every 6 trials, and the decay of the fairness (proportion of pot) of offers is in equation (1). For study 4 we included an additional equation (2) on line 309 which specified how offers were generated. As a result of this pot sizes were scaled to fit with those equations and the rule about reward. Beyond this, there were no other formulas used for pot size creation. Moreover, fairness decays are shown in Figure 1. We have now re-written the section on pot sizes in the methods (Lines 207 - 211).

Statistical models / Computational modeling

9) It would improve clarity to specify all linear mixed effects models

Response: We apologise to the reviewer, although we had included all equations for the LMMs for studies 1-3 (See equation 3), we note that we had omitted two of the LMMs for study 4 that added in the additional measures of loneliness and depression. We have now added this to the methods (equations 4 and 5, lines 378 and 379).

10) 332-333: There is again this one to one mapping between reward and fairness and this has not been well introduced. I understand that the computational framework for this were DDMs but I would still think that it is important to check whether they outperform simple fixed strategies or the linear mixed models. A formal model comparison should test this

Response: The reviewer is correct that model comparisons are an important component of ensuring a model is accurate in accounting for data. For this reason we included a formal model comparison (figure 4B) to compare a model that has a fixed strategy where none of the experimental variables influenced evidence accumulation (the standard DDM), one in which evidence accumulation was influenced by the value of the rewards being offered (the reward DDM) and one where the fairness of offers influenced the evidence accumulation (fairness DDM). We could have performed a DDM with a completely fixed strategy across environments, however this would not be consistent with our goal to explain the behavioural effects we observed between environments. Moreover, we show that the winning model was the only one that could produce the same pattern of results that were actually observed (figure 4C and Supplementary figure 2 and table below).

We did not think it is appropriate to compare LLMs and DDMs with each other. This is because the LLM are looking for effects of experimentally manipulated variables on the dependent variable. In contrast DDMs are process models that are trying to understand the mechanisms that might underlie the distributions of leaving times in the experiment. In addition to this, the algorithms used to fit the parameters of the models to the data are completely different, and thus not easily comparable to each. We would also note that this is not common practice to compare DDMs to LLMs for these reasons. In addition to the formal comparison we also highlighted that through simulations the fixed DDM and the other two

models. In the revised manuscript we have highlighted our rationale for model comparison and for the use of DDMs in more detail (Lines 439 - 453).

11) The starting bias for the DDMs is not well explained. An additional equation might help here

Response: We apologise to the reviewer and thank them for spotting this. The starting bias which is commonly deployed in DDMs is now included in the equation (equation 7), as well as being highlighted in figure 4 and explained in the text below the equation (Lines 412 - 415).

12) I am missing random effects comparisons in the computational modeling section as well as parameter and model recovery.

Response: The reviewer is correct in highlighting that validating the models is important. We are not sure exactly what the reviewer is referring to in regards to model comparison approaches here. As highlighted above we completed a formal model comparison to test our fairness DDM model compared to alternatives, and it was better in explaining the data on all metrics (BIC, proportion of participants the model wins on BIC, and simulation of the data). In line with many studies we did not use a hierarchical approach to create a random effects structure, as they are not always well configured in tasks which do not have typical distributions of the parameters. This is the case in this paradigm where leaving times do not conform to the typical distributions found in reaction time based experiments of cognitive or choice tasks that DDMs are typically fitted too. This is because reaction times here are anywhere from 1s up to over 100s, whereas most experiments look at RTs in the range of less than 10s.

We do however in supplementary figure 3 show that the vast majority of participants (65%) are best explained by the Fairness DDM, thus almost any metric of model fitting and comparison will show that this model is the best fitting. To ensure our modelling was effective and for transparency we have plotted the distributions of parameters (supplementary figure 4), provided the results of the simulated leaving times of the three models included in the formal model comparison (figure 4) and provided the results of mixed-model statistics performed on the simulated data from the fitted models to show only the fairness DDM can replicate the patterns of behaviour observed in study 4 and does statistically (See Results section '*A fairness-adapted evidence accumulation model accounts for decisions to leave*', Figure 4, supplementary table 6). For these reasons we believe we have shown convincing evidence that the model can account for the data. We have now rewritten parts of the methods section to make our formal model comparison approach clear (Lines 439 - 453).

7th Feb 24

Dear Dr Gabay,

Thank you for your patience during the peer-review process. Your manuscript titled "Time to leave: Computations of when to end a social interaction depend on social environment-shaped opportunity-costs" has now been seen by 3 reviewers, and I include their comments at the end of this message. Whilst Reviewers #1 and #2 are happy with your revisions, Reviewer #3 raised some outstanding concerns, which we hope you can address in some final revisions.

We therefore invite you to revise and resubmit your manuscript, along with a point-by-point response to the reviewers. Please highlight all changes in the manuscript text file.

In particular, Reviewer #3 raises some concerns about the statistical analysis of your work. We ask you to address issue #2 through suitable revisions and to provide a detailed response to issue #1.

I am attaching an Editorial Requests Table that details critical reporting requirements for the revised manuscript. Please attend to each item and ensure your manuscript is fully compliant. We are requesting that your manuscript aligns with these requirements to facilitate further processing and potential acceptance of the revision. If your revised manuscript is not aligned with these requests on major issues, such as those concerning statistics, it may be returned to you for further revisions without re-review. Additional information can be found in our style and formatting guide Communications Psychology formatting guide.

Please use the following link to submit your

- revised manuscript,
- point-by-point response to the referees' comments,
- cover letter (as a separate document),
- the Editorial Policy Checklist (see below),
- the Reporting Summary (see below), and
- the completed Editorial Request Table (attached):

[link redacted]

We hope to receive your revised paper within 6 weeks; please let us know if you aren't able to submit it within this time so that we can discuss how best to proceed. If we don't hear from you, and the revision process takes significantly longer, we may close your file. In this event, we will still be happy to reconsider your paper at a later date, provided it still presents a significant contribution to the literature at that stage.

We would appreciate it if you could keep us informed about an estimated timescale for

resubmission, to facilitate our planning.

Best regards,

Antonia Eisenkoeck

Antonia Eisenkoeck
Senior Editor
Communications Psychology

REVIEWER REPORTS:

Reviewer #1 (Remarks to the Author):

I thank the authors for their responses and the hard work that has gone into the revision - I'm happy to recommend to accept for publication

Reviewer #2 (Remarks to the Author):

The authors have done an excellent job of addressing all comments raised in my review. I have no further comments.

Reviewer #3 (Remarks to the Author):

The authors have answered most of my concerns about the paper, for instance reframing the introduction and strengthening the links between ecological and social decision making in the introduction but two important concerns remain:

1) Testing whether the DDM framework is best at capturing behavior in their study. I understand that they used the DDM framework to model participants' choices but it is an open question whether simpler models with fixed parameters are better equipped. These DDMs have 3 parameters and I would want to see that simple linear models are not better than the DDM model. Authors should check this by estimating their effects of reward, fairness etc with simple linear regressions per participant with slopes and intercepts as free parameters. Then they could compare the BICs of these simpler models with less parameters to the DDMs to test whether the more complex DDMs are in fact necessary and best suited to explain task behavior.

2) I am still unsure about the simulated data that was compared to the real task data: is this the data that is described in the sample size justification? Particularly if random effect model comparisons are not feasible, I would want to see posterior predictive checks to interpret the robustness of the the

modeling results. Authors should simulate task data with the DDM equations with participants' parameters and then compare these data to the participants' real responses. They should also test whether they can recover the model parameters of the simulated data.

EDITORIAL POLICIES

We ask that you ensure your manuscript complies with our editorial policies and reporting requirements.

To that end, we require revised manuscripts to be accompanied by two completed items: a reporting summary that collects information on study design and procedure, and an editorial policy checklist that verifies compliance with all required editorial policies.

- Nature Research Reporting Summary
- Editorial Policy Checklist

All points on the policy checklist must be addressed. Your revised manuscript can only be sent back to the referees if these checklists are completed and uploaded with the revision.

Notes: If you have submitted a Stage 1 Registered Report, Review, Primer, Comment, or Perspective you do not need to submit these forms. If you have already submitted these forms, you may disregard this request.

* TRANSPARENT PEER REVIEW: Communications Psychology uses a transparent peer review system. This means that we publish the editorial decision letters including Reviewers' comments to the authors and the author rebuttal letters online as a supplementary peer review file. However, on author request, confidential information and data can be removed from the published reviewer reports and rebuttal letters prior to publication. If your manuscript has been previously reviewed at another journal, those Reviewers' comments would not form part of the published peer review file.

Communications Psychology is committed to improving transparency in authorship. As part of our efforts in this direction, we are now requesting that all authors identified as 'corresponding author'

create and link their Open Researcher and Contributor Identifier (ORCID) with their account on the Manuscript Tracking System prior to acceptance. ORCID helps the scientific community achieve unambiguous attribution of all scholarly contributions. You can create and link your ORCID from the home page of the Manuscript Tracking System by clicking on 'Modify my Springer Nature account' and following the instructions in the link below. Please also inform all co-authors that they can add their ORCIDs to their accounts and that they must do so prior to acceptance.

If you experience problems in linking your ORCID, please contact the Platform Support Helpdesk.

Reviewer #1 (Remarks to the Author):

I thank the authors for their responses and the hard work that has gone into the revision - I'm happy to recommend to accept for publication

Response: We thank the reviewer for their helpful review and welcome the recommendation.

Reviewer #2 (Remarks to the Author):

The authors have done an excellent job of addressing all comments raised in my review. I have no further comments.

Response: We thank the reviewer for their thoughtful review and pleased we have addressed all their comments.

Reviewer #3 (Remarks to the Author):

The authors have answered most of my concerns about the paper, for instance reframing the introduction and strengthening the links between ecological and social decision making in the introduction but two important concerns remain:

Response: We are pleased to see we have addressed most of the reviewer's concerns. We respond to their two remaining comments below. We would like to note that their concerns do not relate to many aspects of our results, as the modelling is only undertaken on the final study. Thus, the key findings of our manuscript stand, regardless of any specific concerns about particular features of the computational models.

1) Testing whether the DDM framework is best at capturing behavior in their study. I understand that they used the DDM framework to model participants' choices but it is an open question whether simpler models with fixed parameters are better equipped. These DDMs have 3 parameters and I would want to see that simple linear models are not better than the DDM model. Authors should check this by estimating their effects of reward, fairness etc with simple linear regressions per participant with slopes and intercepts as free parameters. Then they could compare the BICs of these simpler models with less parameters to the DDMs to test whether the more complex DDMs are in fact necessary and best suited to explain task behavior.

Response: We agree with the reviewer that testing whether computational models are good fits to the data is a principled approach to understanding their ability to explain choice behaviour. It is for this reason that we did compare the three different versions of a DDM computational model. However, within our DDMs fixing parameters would not be a valid approach. We outline why below.

The purpose of fixing model parameters is typically to show that a particular feature of a model, that putatively maps onto a specific cognitive/neural process, is or isn't influencing

behaviour in a task. Fixing parameters is therefore informative in many models of choice behaviour (Lockwood & Klein-Flugge, 2020, *SCAN*). For instance in a rescorla-wagner reinforcement learning model for a two-armed bandit task fixing the learning rate parameter to 1 means that each outcome completely updates the expectation on the next trial, and as such people are operating a fixed choice rule and not integrating outcomes over time (i.e. they don't show a probabilistic or value-based learning effect). However, it is not the case that fixing model parameters always has a simple interpretation or makes logical sense. For instance, for a typical effort-discounting computational model in a task where people make decisions of whether to exert different levels of effort for different rewards, fixing the parameter to a certain level (e.g. 1) would just mean that all participants discounted rewards by effort to the same degree (Lockwood et al., 2017, *Nature Human Behaviour*), not that people did not discount rewards by effort. The putative cognitive process is therefore present in all people and does not vary i.e. it is quite different from the interpretation

For evidence accumulation models fixing parameters does not have the same type of interpretation as the RL example above, nor is it as trivial to complete for the following reasons:

(i) DDMs are theoretically informed models that putatively map on to the set of processes in the brain that make decisions. Removing parameters therefore breaks the theoretical link between the DDM and what using it is supposed to imply about the decision-making processes taking place. The notion is that evidence for a decision starting at a particular level is driven by a certain level of starting activity (starting bias), that the evidence/activity associated with decision increases over time at a certain rate (drift rate) until it reaches a certain level (threshold) that triggers a choice. Fixing any one of these parameters means that it could not be considered a DDM and thus would mean its interpretation would not make theoretical sense within a DDM framework.

(ii) As DDMs are trying to model distributions of reaction times (or in this case leaving times), rather than a changing value, fixing or removing one parameter does not remove its influence, but fundamentally changes what is being estimated by the model. For instance removing the threshold would mean that the model never makes a decision to leave. Or fixing it, would just mean that every decision is reached with the same amount of evidence. Interpreting what this means would be challenging.

(iii) What value should the parameters be fixed too? Unlike RL models, the parameter ranges and values for DDMs are somewhat arbitrary, as they depend upon the underlying task, the means and SDs of observed reaction times. This means setting the parameter to any fixed value a priori would be challenging, especially as RTs in this experiment were not in the typical range of most experiments using DDM which examine RTs that are far shorter. We could possibly now try and fix the parameter post-hoc based on the model parameters that we have observed, but this would not be principled as - like many post-hoc analyses approaches - it can be biased to find a result one direction or the other based on a choice which is not independent of the data. It would also be unclear which model we should take that value from.

We therefore feel that it is not necessary nor principled to undertake the proposed analyses. In the revised manuscript we now include a further explanation of why we did not include further model comparison analyses.

2) I am still unsure about the simulated data that was compared to the real task data: is this the data that is described in the sample size justification? Particularly if random effect model comparisons are not feasible, I would want to see posterior predictive checks to interpret the robustness of the the modeling results. Authors should simulate task data with the DDM equations with participants' parameters and then compare these data to the participants' real responses. They should also test whether they can recover the model parameters of the simulated data

Response: We thank the reviewer for helping us realise the manuscript may not have clearly spelled out how we had already performed posterior predictive checks on the data. We agree that posterior predictive checks are important, and we had performed the analyses the reviewer suggested. In the revised manuscript we now highlight more clearly how we had two simulation approaches across the study, one set of simulations performed to estimate the required sample size in statistical models, and a separate set of simulations of the DDMs. Specifically:

Methods Line 445:

- We performed simulations based on the parameters estimated on subjects data (these are separate simulations from those used to estimate sample sizes above). For each set of parameters, we generated LT distributions by running 1,000 simulations of the model (that is, by producing this number of decision trajectories using equation (6) for each environment condition). To further assess the quality of the fits resulting from the best set of participant-specific parameters (those that maximised the LL function in equation (11)), we computed correlation coefficients between the average LT from the data and the model for all participants and conditions. We then performed the same statistics performed in the main behavioural analyses of study 4's data on the simulated data from each model.

Figure 4c:

- The simulations of each model are shown in Figure 4c. Only one model - the fairness DDM (top right panel) - can show the two main effects (partner fairness and environment generosity) that are shown in the behavioural data (top left panel), the lower panels show the other model predictions and neither looks at all similar to the top left panel showing the real data.

From line 655 in the main text:

- When running a linear mixed-effects model on the simulated data from the winning Fairness-DDM, we found main effects of partner-type and social environment, (partner-type: $b = -6.54$, $SE = 0.13$, $t(161900) = -49.827$, $p < 0.001$; $X^2(1) = 3877.81$, $p < 0.001$; environment: $b = 1.66$, $SE = 0.13$, $t(161900) = 12.734$, $p < 0.001$; $X^2(1) = 688.65$, $p = 0.001$; see Supplementary results and Supplementary Table 4 for the other models).

In addition supplementary table 4:

- We report the same statistical analyses for the losing models, and show that statistically these models cannot replicate the main results in the behavioural data, indicating that only the winning model provides a good account of the data.

Line 653:

- To further assess the quality of fit of the winning model, we computed correlation coefficients between the average leaving time from the data and the winning model for all participants and for each environment. The Spearman's rho was 0.98 and 0.99 for the high and low generosity environments, respectively, indicating good fit.

14th Mar 24

Dear Dr Gabay,

Your manuscript titled "Social environment-based opportunity costs dictate when people leave social interactions". I am delighted to say that we are happy, in principle, to publish a suitably revised version in Communications Psychology under the open access CC BY license (Creative Commons Attribution v4.0 International License).

We therefore invite you to edit your manuscript to comply with our format requirements and to maximise the accessibility and therefore the impact of your work.

Overall, the manuscript is in very good shape and mostly compliant with our guidelines. Thank you for the careful preparation of your files. I have tried to minimize the items on the attached Editorial Request Table to not list too many requirements that you already meet. However, this means in return that I ask you to run potential non-requested changes by me before you implement these, to avoid accidental non-compliance with our formatting requirements that will delay acceptance.

SUBMISSION INFORMATION:

OPEN ACCESS:

Communications Psychology is a fully open access journal. Articles are made freely accessible on publication under a CC BY license (Creative Commons Attribution 4.0 International License). This license allows maximum dissemination and re-use of open access materials and is preferred by many research funding bodies.

For further information about article processing charges, open access funding, and advice and support from Nature Research, please visit <https://www.nature.com/commspsychol/article-processing-charges>

At acceptance, you will be provided with instructions for completing this CC BY license on behalf of all authors. This grants us the necessary permissions to publish your paper. Additionally, you will be asked to declare that all required third party permissions have been obtained, and to provide billing information in order to pay the article-processing charge (APC).

* TRANSPARENT PEER REVIEW: Communications Psychology uses a transparent peer review system.

On author request, confidential information and data can be removed from the published reviewer reports and rebuttal letters prior to publication. If you are concerned about the release of confidential data, please let us know specifically what information you would like to have removed. Please note that we cannot incorporate redactions for any other reasons.

* CODE AVAILABILITY: All Communications Psychology manuscripts must include a section titled "Code Availability" at the end of the methods section. We require that the custom analysis code supporting your conclusions is made available in a publicly accessible repository at this stage; please choose a repository that generates a digital object identifier (DOI) for the code; the link to the repository and the DOI must be included in the Code Availability statement. Publication as Supplementary Information will not suffice.

* DATA AVAILABILITY:

[link redacted]

Best wishes,

Marike

Marike Schiffer, PhD
Chief Editor
Communications Psychology